# Gradual Transition from Bellman Optimality Operator to Bellman Operator in Online Reinforcement Learning

Motoki Omura [1]   Kazuki Ota [1]   Takayuki Osa [2]   Yusuke Mukuta [1,2]   Tatsuya Harada [1,2]

## Abstract

For continuous action spaces, actor-critic methods are widely used in online reinforcement learning (RL). However, unlike RL algorithms for discrete actions, which generally model the optimal value function using the Bellman optimality operator, RL algorithms for continuous actions typically model Q-values for the current policy using the Bellman operator. These algorithms for continuous actions rely exclusively on policy updates for improvement, which often results in low sample efficiency. This study examines the effectiveness of incorporating the Bellman optimality operator into actor-critic frameworks. Experiments in a simple environment show that modeling optimal values accelerates learning but leads to overestimation bias. To address this, we propose an annealing approach that gradually transitions from the Bellman optimality operator to the Bellman operator, thereby accelerating learning while mitigating bias. Our method, combined with TD3 and SAC, significantly outperforms existing approaches across various locomotion and manipulation tasks, demonstrating improved performance and robustness to hyperparameters related to optimality. The code for this study is available at https://github.com/motokiomura/annealed-q-learning.

## 1. Introduction

In recent years, online reinforcement learning (RL) has been widely applied to robotics (Schulman et al., 2015; Haarnoja et al., 2018) and gaming (Mnih et al., 2013; 2015; Badia et al., 2020; Tessler et al., 2017), demonstrating remarkable performance, particularly in tasks with discrete action spaces like games. Q-learning-based algorithms, commonly

used for discrete action tasks, estimate the optimal Q-value via the Bellman optimality operator. However, in tasks with continuous action spaces, computing $\max_{a'} Q(s', a')$ for an infinite number of actions is challenging. Actor-critic-based algorithms address this by estimating the Q-value for the current policy using the Bellman operator. In these cases, policy improvement is achieved solely through policy updates, leading to slower performance improvement and reduced sample efficiency (Ji et al., 2024). In tasks with continuous action spaces, such as robotic control, sample collection is costly, which makes low sample efficiency a critical challenge. Although some prior studies have explored modeling (soft) optimal values instead of the values of the current policy, they faced challenges such as increased computational cost and instability (Haarnoja et al., 2017; Garg et al., 2023; Kalashnikov et al., 2018).

This study first examines the effectiveness of modeling optimal values in actor-critic methods for online RL. Preliminary experiments were conducted with a low-dimensional environment using a tabular actor-critic method. These experiments compared SARSA-based updates, which used the Bellman operator, to Q-learning-based updates, which used the Bellman optimality operator, for critic learning. The results showed that in Q-learning-based methods, the Q-value directly approached the optimal value, whereas in SARSA-based updates, the Q-value improved only after the policy improved. This delay in policy learning caused the Q-value to converge to its optimal value more slowly in the SARSA-based approach.

This finding motivates the application of the Bellman optimality operator in actor-critic methods. However, using the Bellman optimality operator with function approximators to model Q-values introduces overestimation bias due to the stochasticity of Q-values and the max operator in target values (Thrun & Schwartz, 1993; Lan et al., 2020; Chen et al., 2021). Simulating this stochasticity by injecting noise into Q-values in our preliminary experiments showed that Q-learning-based methods converged to values larger than the optimal Q-value due to overestimation bias. Conversely, SARSA-based methods, which do not involve a max operator, exhibited lower sensitivity to noise.

In summary, while the Bellman optimality operator acceler-

---

[1]The University of Tokyo [2]RIKEN. Correspondence to: Motoki Omura <omura@mi.t.u-tokyo.ac.jp>.

*Proceedings of the 42nd International Conference on Machine Learning*, Vancouver, Canada. PMLR 267, 2025. Copyright 2025 by the author(s).

ates learning, it also increases overestimation bias, leading to convergence to inflated Q-values. Conversely, the Bellman operator suppresses overestimation bias but results in slower learning.

To address this, we propose a gradual transition from the Bellman optimality operator to the Bellman operator. This approach accelerates learning while eventually estimating values with reduced bias. In the preliminary experiments, linearly interpolating target values calculated with each operator, combined with annealing the degree of optimality, resulted in faster learning and less biased estimation.

This method has additional benefits. While overestimation bias may occur in the early stages of learning depending on the annealing schedule, it can promote exploration, which is advantageous at this stage (Lan et al., 2020). In the later stages of learning, the current policy is expected to converge, making it undesirable to introduce bias for further policy improvement. Therefore, applying updates based on the Bellman operator, which inherently reduces bias, becomes a reasonable approach.

In tasks with continuous action spaces, which we focus on in this study, the Bellman optimality operator is computationally intractable due to the max operation. To address this, we use the expectile loss (Kostrikov et al., 2022), which enables operations analogous to the max operator in continuous action spaces and a smooth interpolation between the Bellman optimality operator and the Bellman operator. By gradually reducing a parameter $\tau$ representing the estimated expectile from a value close to 1 to 0.5, our operator transitions smoothly from the Bellman optimality operator to the Bellman operator. This approach can be implemented by replacing the L2 loss used for critic learning in existing actor-critic methods with the expectile loss and annealing the parameter $\tau$ related to optimality.

Experiments on various locomotion and manipulation tasks in continuous action spaces tested the proposed Annealed Q-learning (AQ-L) method combined with TD3 and SAC. The proposed methods significantly outperformed widely used algorithms such as TD3 and SAC. Additionally, annealing improved robustness to hyperparameters related to optimality and enhanced performance compared to static optimality estimates.

The contributions of this study are as follows:

- Preliminary experiments provided the insight that in actor-critic methods, using the Bellman optimality operator accelerates learning but introduces overestimation bias, while the Bellman operator reduces bias at the cost of slower learning.

- We proposed a method for annealing from the Bellman optimality operator to the Bellman operator, enabling

faster learning with reduced bias. This method can be easily implemented in existing actor-critic methods using the expectile loss.

- We demonstrated that Annealed Q-learning combined with TD3 and SAC achieves significantly better performance than widely used algorithms in continuous action tasks, while annealing improves performance and robustness to optimality-related hyperparameters.

## 2. Preliminaries

### 2.1. Reinforcement Learning

In RL, the problem is defined within a Markov decision process (MDP) framework presented by the tuple $(\mathcal{S}, \mathcal{A}, \mathcal{P}, r, \gamma, d)$. Here, $\mathcal{S}$ denotes the set of all possible states, $\mathcal{A}$ denotes the set of all possible actions, $\mathcal{P}(s_{t+1}|s_t, a_t)$ is the transition probability from one state to another given a specific action, $r(s, a)$ represents the reward function assigning values to each state-action pair, $\gamma$ is the discount factor that diminishes the value of future rewards, and $d(s_0)$ is the probability distribution of initial states. The policy $\pi(a \mid s)$ is the probability of taking a specific action in a given state. The objective of RL is to discover a policy that maximizes the expected sum of discounted rewards, denoted as $\mathbb{E}[R_0 \mid \pi]$, where $R_t$ is the return calculated as $R_t = \sum_{k=t}^{T} \gamma^{k-t} r(s_k, a_k)$ and $T$ is a task horizon.

### 2.2. Bellman Operator and Bellman Optimality Operator

In RL, the Bellman (expectation) operator and Bellman optimality operator (Sutton, 1988) play fundamental roles in defining the iterative updates for value functions in MDPs. Here, we describe the update of the action-value function $Q(s, a)$ instead of the state-value function $V(s)$.

**Bellman Operator** For a given policy $\pi$, the Bellman operator $\mathcal{T}^\pi$ is defined on the Q-function $Q^\pi(s, a)$ as:

$$\mathcal{T}^\pi Q(s, a) = \mathbb{E}_{s' \sim \mathcal{P}} \left[ r(s, a) + \gamma \mathbb{E}_{a' \sim \pi} Q(s', a') \right]$$

Applying $\mathcal{T}^\pi$ repeatedly leads to the Q-function satisfying the Bellman equation: $Q^\pi(s, a) = \mathcal{T}^\pi Q^\pi(s, a)$.

The value function in SARSA (Rummery & Niranjan, 1994) and the critic in actor-critic methods are generally updated based on the Bellman operator $\mathcal{T}^\pi$. These algorithms follow the policy iteration framework, where policy evaluation is performed using updates based on $\mathcal{T}^\pi$, followed by policy improvement using the updated value function to obtain a better policy. In other words, to derive an improved policy, the value function must accurately evaluate the current policy (Sutton & Barto, 2018).

**Bellman Optimality Operator** The Bellman optimality operator $\mathcal{T}^*$ is defined to obtain the optimal Q-function $Q^*$, assuming the agent selects the action that maximizes expected future rewards:

$$\mathcal{T}^* Q(s,a) = \mathbb{E}_{s' \sim \mathcal{P}}\left[r(s,a) + \gamma \max_{a'} Q(s',a')\right].$$

This operator is contractive, ensuring that iterative applications converge to the optimal action-value function $Q^*$, and lead to the Bellman optimality equation: $Q^*(s,a) = \mathcal{T}^* Q^*(s,a)$.

In methods such as Q-learning (Watkins, 1989) and DQN (Mnih et al., 2013; 2015), the action-value function is updated directly toward the optimal action-value function using the Bellman optimality operator $\mathcal{T}^*$.

### 2.3. Expectile Loss

The expectile loss is an asymmetric loss function with a parameter $\tau$, and minimizing this loss enables the estimation of the expectile $\tau$ of the given data. When $\tau = 0.5$, the loss function is equivalent to the L2 loss, allowing the estimation of the mean. As $\tau$ approaches 1, it estimates values closer to the maximum. This loss function is used in offline RL to compute the Bellman optimality operator $\mathcal{T}^*$, and the corresponding method is known as Implicit Q-learning (IQL) (Kostrikov et al., 2022).

IQL is an algorithm widely used in offline RL to learn the value function and the policy using only actions from the offline dataset, without relying on actions from the policy being learned. IQL can learn near-optimal value functions in an in-distribution manner. This is achieved by considering the target value of the value function as a random variable depending on the action and estimating the upper expectile, e.g., $\tau = 0.9$, with the expectile loss. Thus, the loss for learning the Q-function is as follows:

$$L(\theta) = \mathbb{E}_{(s,a,s',a') \sim \mathcal{D}}[L_2^\tau(r(s,a) + \gamma Q_{\bar\theta}(s',a') \\ - Q_\theta(s,a))], \quad (1)$$

where $L_2^\tau(u) = |\tau - \mathbb{1}(u < 0)|u^2$, $Q_\theta$ is an approximated Q-function parameterized by $\theta$, and $Q_{\bar\theta}$ is the target network with parameters $\bar\theta$.

In a stochastic environment, not only the randomness of the action but also the randomness of the state transition affects the target value, so the following loss function involving the V-function is used:

$$L(\psi) = \mathbb{E}_{(s,a) \sim \mathcal{D}}[L_2^\tau(Q_{\bar\theta}(s,a) - V_\psi(s))], \\ L(\theta) = \mathbb{E}_{(s,a,s') \sim \mathcal{D}}[(r(s,a) + \gamma V_\psi(s') - Q_\theta(s,a))^2]. \quad (2)$$

Thus, IQL computed the Bellman optimality operator's maximum over actions using only actions from the dataset.

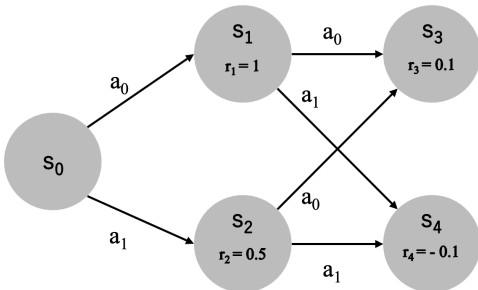

*Figure 1.* A simple MDP used in preliminary experiments. A reward $r_i$ is obtained upon reaching state $s_i$, and the episode terminates when the agent reaches either state $s_3$ or $s_4$. The discount rate $\gamma$ is set to 0.9.

## 3. Gradual Transition from Bellman Optimality Operator to Bellman Operator

In this section, we first discuss the effectiveness of the Bellman optimality operator $\mathcal{T}^*$ and the Bellman operator $\mathcal{T}^\pi$ in actor-critic methods by employing Q-learning-based and SARSA-based updates, respectively. Our preliminary experiments demonstrate that Q-learning-based updates, which further enhance improvement in the critic, accelerate learning but may lead to overestimation bias. Based on these findings, we propose an annealing approach that gradually transitions from Q-learning-based to SARSA-based updates, i.e., from the Bellman optimality operator $\mathcal{T}^*$ to the Bellman operator $\mathcal{T}^\pi$, by reducing the influence of the max operator, thereby mitigating the residual bias in the later stages of training.

### 3.1. Bellman Optimality Operator Accelerates Learning

Many commonly used algorithms for continuous action tasks in online RL (Fujimoto et al., 2018; Haarnoja et al., 2018; Schulman et al., 2017) update the critic with SARSA-based updates under the Bellman operator $\mathcal{T}^\pi$. The value function estimates the value under the current policy, relying solely on policy updates for policy improvement. Replacing the critic's update mechanism with a Q-learning-based approach under $\mathcal{T}^*$ may accelerate policy improvement and improve learning efficiency. To investigate this, we conducted preliminary experiments in a simple environment, as shown in Figure 1. In this experiment, we evaluated how many steps it took for the learned Q-values to approach the optimal values in the environment. We used a conventional SARSA-based actor-critic method and a Q-learning-based actor-critic method for learning algorithms. In the SARSA-based method, Q-values and policy logits $\theta_{s,a}$ were stored in a table, and the Q-values were updated in a SARSA-like manner ($Q(s,a) \leftarrow Q(s,a) + \alpha(r + \gamma \mathbb{E}_{a' \sim \pi}[Q(s',a')] -$

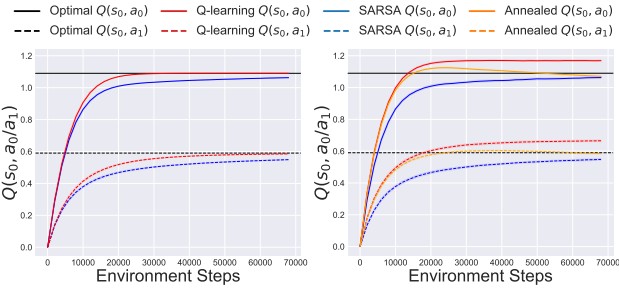

*Figure 2.* **Left**: The estimated values of $Q(s_0, a_0)$ and $Q(s_0, a_1)$ when using tabular actor-critic with a Q-learning-based critic update under $\mathcal{T}^*$, compared to a SARSA-based critic update based on $\mathcal{T}^\pi$ in the environment of Figure 1. **Right**: The estimated Q-values when Gaussian noise from $\mathcal{N}(0, 0.3)$ is added to the Q-value before calculating the target value to simulate the randomness in Q-values when approximated with functions. In *Annealed*, both Q-learning-based and SARSA-based target values are computed, and the learning process transitions linearly from Q-learning-based at the beginning to SARSA-based at the end.

$Q(s, a))$), while the policy was updated using a policy gradient method ($\theta_{s,a} \leftarrow \theta_{s,a} + \alpha \nabla_\theta \log \pi_\theta(a|s) Q(s, a)$), where $\alpha$ represents the step size. In the Q-learning-based method, Q-values were updated in a Q-learning-like manner ($Q(s, a) \leftarrow Q(s, a) + \alpha(r + \gamma \max_{a'} Q(s', a') - Q(s, a))$), and the policy was updated in the same manner using the policy gradient method.

The Q-values learned in this experiment are shown in the left panel of Figure 2. The Q-learning-based approach was able to estimate the optimal Q-values faster than the SARSA-based approach. In the SARSA-based method, the policy must first improve before Q-values can move toward the optimal value function, causing a delay compared to the Q-learning-based method.

### 3.2. Bellman Operator is Less Biased

As demonstrated, Q-learning-based updates in actor-critic methods can accelerate learning compared to SARSA-based updates. However, calculating the target value using the max operation in Q-learning-based methods can introduce overestimation bias. When representing the Q-values with a function approximator, the approximation errors and the stochasticity of gradient methods result in noisy estimated Q-values. Computing the maximum of these noisy Q-values leads to overestimation. This can be explained as follows: the ideal target value is $r + \gamma \max_{a'} \mathbb{E}[Q(s', a')]$, but the computable target value is $r + \gamma \mathbb{E}[\max_{a'} Q(s', a')]$. According to Jensen's inequality, $r + \gamma \mathbb{E}[\max_{a'} Q(s', a')] \geq r + \gamma \max_{a'} \mathbb{E}[Q(s', a')]$. Thrun & Schwartz (1993); Lan et al. (2020); Chen et al. (2021) provided more specific definitions of bias. They assumed that the Q-values are equal for all actions, corresponding to the scenario where overestima-

tion is maximized. They expressed the randomness of the Q-values by adding noise $\epsilon_{s,a}$. The bias is then defined as: $Z = r + \gamma \max_{a'}(Q(s', a') + \epsilon_{s',a'}) - (r + \gamma \max_{a'} Q(s', a'))$. Even when the mean of $\epsilon_{s',a'}$ is 0, Jensen's inequality shows:

$$\mathbb{E}_\epsilon[Z] = \gamma \mathbb{E}[\max_{a'} \epsilon_{s',a'}] \geq \gamma \max_{a'} \mathbb{E}[\epsilon_{s',a'}] = 0, \quad (3)$$

indicating overestimation occurs.

To simulate the overestimation bias due to the randomness, we conducted an experiment in which noise was added to Q-values when calculating the target value as $r + \gamma \max_{a'}(Q(s', a') + \epsilon_{s',a'})$ for Q-learning and $r + \gamma \mathbb{E}_{a' \sim \pi}[Q(s', a') + \epsilon_{s',a'}]$ for SARSA, where $\epsilon_{s',a'}$ was sampled from a normal distribution with mean 0. The results are shown in the right panel of Figure 2. In the Q-learning-based method, convergence was faster, but due to the noise, the estimated Q-values converged to values larger than the optimal ones, confirming overestimation. In the SARSA-based case, the expected value is used instead of the maximum value, and since the mean of $\epsilon_{s',a'}$ is 0, its impact is minimal.

### 3.3. Gradual transition from Q-learning to SARSA

The results presented in the previous section illustrate the advantages and disadvantages of SARSA-based and Q-learning-based updates in actor-critic methods. Q-learning-based updates accelerate learning but lead to convergence to higher values, whereas SARSA-based updates exhibit slower learning but reduced bias. To leverage the strengths of both approaches, we propose a method that initially employs Q-learning-based updates and gradually transitions to SARSA-based updates as learning progresses. Specifically, in our preliminary experiments, we compute both the target values of Q-learning and SARSA, given by $Q_{\text{QL}} = r + \gamma \max_{a'} Q(s', a')$ and $Q_{\text{SARSA}} = r + \gamma \mathbb{E}_{a' \sim \pi}[Q(s', a')]$, and use a weighted average as the target value:

$$Q_{\text{AQ-L}} = w Q_{\text{QL}} + (1 - w) Q_{\text{SARSA}}, \quad (4)$$

where $w$ is a weight that linearly decays from 1 to 0. The estimated Q-value is updated as follows:

$$Q(s, a) \leftarrow Q(s, a) + \alpha(Q_{\text{AQ-L}} - Q(s, a)). \quad (5)$$

The results, as represented by the orange curve in Figure 2, demonstrate fast and less biased Q-value estimations. The settings of these preliminary experiments and other details are provided in Appendix A.

### 3.4. Annealed Q-learning: Gradual Transition of Operator in Continuous Action Spaces

In the previous section, we showed that estimating the optimal Q-value through the max operation can accelerate

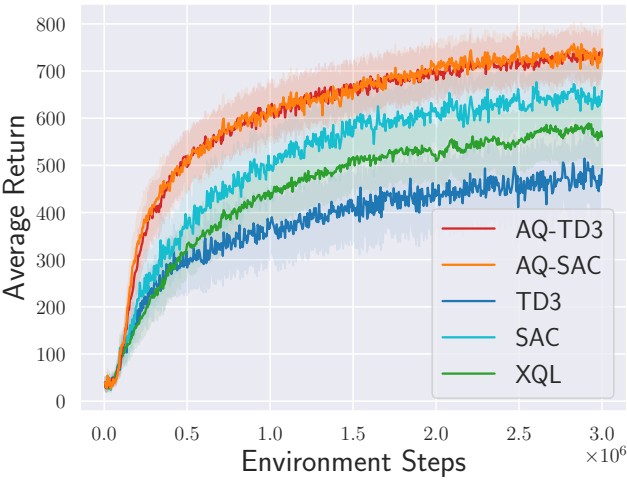

*Figure 3.* The average scores across the 10 locomotion tasks in DM Control.

learning and that annealing this optimality allows for efficient estimation of less biased values in the final stage. However, in continuous action tasks, it is generally not feasible to directly compute the maximum Q-value. To address this, we use the expectile loss, as introduced in Kostrikov et al. (2022), which allows for the implicit computation of Q-learning-based target values without employing the max operation. This approach also enables smooth interpolation between Q-learning-based and SARSA-based target values.

In online RL, the expectile loss is used to estimate the optimal value based on actions sampled from the current policy. The loss function for the value function is expressed as follows:

$$L(\theta) = \mathbb{E}_{(s,a,s') \sim \mathcal{D}, a' \sim \pi}[L_2^\tau(r(s,a) + \gamma Q_{\bar{\theta}}(s', a') \\ - Q_\theta(s, a))]. \quad (6)$$

In the expectile loss, setting $\tau = 1$ allows for estimating a Q-learning-based target, while setting $\tau = 0.5$ estimates a SARSA-based target. In this study, we propose a method that begins with a $\tau$ value close to 1 at the start of training and gradually anneals it to 0.5 by the end.

As discussed in the previous section, estimating the Q-learning-based optimal Q-value at the early stages of training and transitioning to estimating the SARSA-based Q-value for the current policy in the later stages enables efficient learning. While overestimation might occur in the early stages depending on the annealing schedule, this can facilitate exploration (Lan et al., 2020). The relationship between this bias and exploration is also discussed in Appendix I. Overestimation increases the chance of the agent selecting overestimated actions, correcting them in the process, and thus broadening the range of actions tried. This bias is particularly beneficial in the early stages of learning when exploration is critical. The SARSA-based updates in

*Table 1.* The average score at 3M steps across the DM Control tasks. The range in parentheses represents the confidence interval.

| Method | Mean | IQM |
|---|---|---|
| AQ-TD3 | **740.3** (731.0 - 749.0) | **820.0** (811.5 - 826.8) |
| AQ-SAC | **746.1** (736.3 - 755.0) | **832.4** (820.5 - 841.6) |
| TD3 | 492.7 (459.5 - 527.1) | 516.0 (461.8 - 571.9) |
| SAC | 657.9 (623.6 - 688.9) | 765.0 (712.8 - 800.9) |
| XQL | 564.4 (522.2 - 604.7) | 628.8 (560.6 - 688.3) |

the later stages of training reduce bias, even though these updates no longer lead to further improvement. By this stage, the policy is expected to have converged; thus, prioritizing bias reduction over further improvement is reasonable.

In the proposed method, $\tau$ is simply linearly decayed over time. While experimenting with several annealing patterns, a linear schedule performed sufficiently well. Specifically, when the initial value of $\tau$ is $\tau_{\text{init}}$, the maximum timestep is $T$, and the current timestep is $t$, then $\tau$ at timestep $t$ can be expressed as:

$$\tau(t) = \tau_{\text{init}} - (\tau_{\text{init}} - 0.5)\frac{t}{T}. \quad (7)$$

We named this method Annealed Q-learning (AQ-L) and conducted experiments using methods combined with TD3 (Fujimoto et al., 2018) and SAC (Haarnoja et al., 2018), referred to as AQ-TD3 and AQ-SAC, respectively. These methods are implemented through a simple modification, where the squared error in the critic update is replaced with the expectile loss, and $\tau$ is annealed as shown in Equation (7).

## 4. Experiments

To verify the effectiveness of the proposed method, we conducted several experiments. Specifically, we examined how the proposed AQ-TD3 and AQ-SAC perform compared to widely used online RL methods such as SAC and TD3. Additionally, we investigated the impact of the annealing of $\tau$ on task scores and the robustness of hyperparameters.

### 4.1. Experimental Setup

we conducted online RL training using 10 challenging locomotion tasks from DM Control (Tassa et al., 2018; Tunyasuvunakool et al., 2020) and 10 difficult manipulation tasks from Meta-World (Yu et al., 2019). Although Meta-World is often used for multi-task learning, this study focuses exclusively on single-task learning. As for the 10 Meta-World tasks, we selected difficult tasks reported to have low success rates in their paper. We employed TD3 and SAC as baseline methods, widely used in continuous action tasks for online RL, along with XQL (Garg et al., 2023). XQL uses

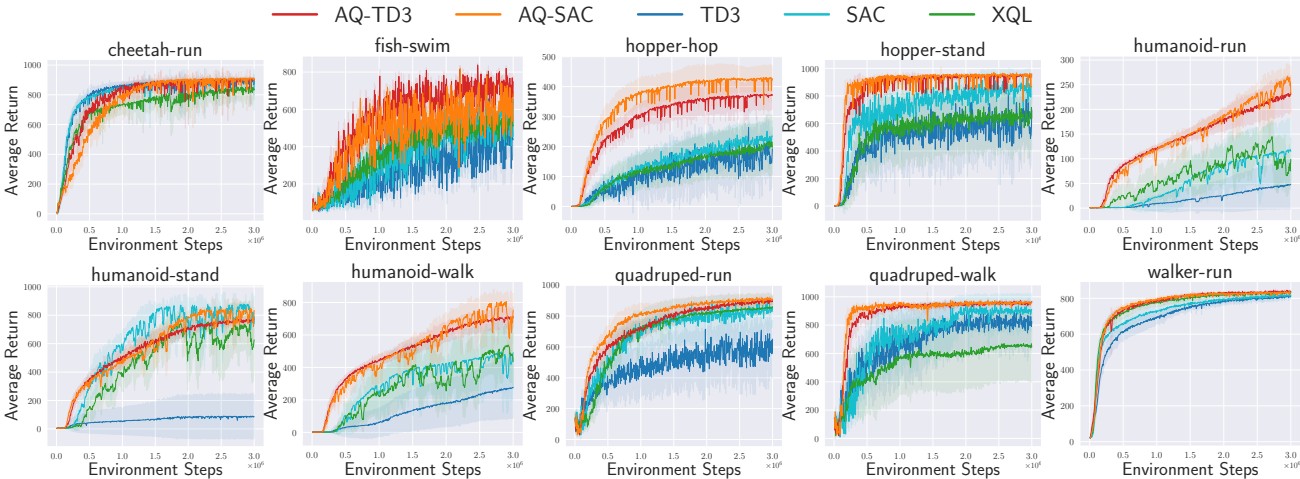

*Figure 4.* The average return for each task in DM Control.

Gumbel regression, assuming a Gumbel distribution for the error distribution, and estimates the soft-optimal value in entropy-maximizing RL, offering a comparison as another method for estimating the optimal value. While Kostrikov et al. (2022) incorporated both a Q-function and a V-function to mitigate performance degradation observed in stochastic environments, AQ-L demonstrated superior performance without employing a V-function, even in extremely stochastic environments. Therefore, we utilize only the Q-function in our approach. Further details on this comparison are provided in Appendix G. The hyperparameters related to annealing are shown in Appendix B. The results for using different annealing durations are presented in Appendix E. For AQ-TD3 and AQ-SAC, all other hyperparameters are kept identical to those of TD3 and SAC, respectively. When $\tau$ is fixed at 0.5, AQ-TD3 and AQ-SAC are precisely the same as TD3 and SAC.

### 4.2. Comparison with Prior Studies

The average scores across 10 tasks from DM Control for AQ-TD3, AQ-SAC and the baseline methods from prior studies are shown in Figure 3. The average scores, interquartile mean (IQM), and their confidence intervals, measured using Agarwal et al. (2021), are listed in Tables 1 and 4. Both AQ-TD3 and AQ-SAC significantly outperformed prior studies. Furthermore, they demonstrated substantial improvements over their respective base algorithms, TD3 and SAC. These results indicate that the simple modifications of employing the expectile loss and annealing $\tau$ in online RL can lead to remarkable performance enhancements. As shown in Table 4, AQ-TD3 and AQ-SAC achieve high scores even with smaller steps, demonstrating a significant improvement in sample efficiency.

The scores for each task are shown in Figure 4. The

maximum return achievable for these tasks is 1000. In hopper-hop, humanoid-run, and humanoid-walk, where existing methods achieved low scores after 3 million training steps, the scores of AQ-TD3 and AQ-SAC increased significantly. In tasks such as hopper-stand, quadruped-run, and quadruped-walk, where existing methods attained consistently high scores, AQ-TD3 and AQ-SAC achieved high scores with fewer steps and exhibited reduced variance, resulting in higher final scores. This suggests improved sample efficiency from optimal value estimation and enhanced stability due to increased exploration early in the training. AQ-TD3 and AQ-SAC also outperformed XQL, another method for estimating the (soft-)optimal value, demonstrating that expectile-based squared loss in online RL is more stable and superior to XQL's exponential-based loss. A straightforward approach to computing the max operation is max-backup, which involves sampling Q-values for multiple actions and using the maximum value. AQ-L outperformed max-backup in terms of both computational efficiency and score. The results and details of this comparison are shown in Appendix D.

Figure 5 shows the average success rate across the manipulation tasks in Meta-World, and the success rates for each task are presented in Figure 6. TD3 exhibited a significantly lower success rate than SAC, suggesting the necessity of exploration enhancement, such as entropy maximization in SAC. Nevertheless, AQ-TD3 achieved a slight performance improvement over TD3, which struggled to learn effectively. AQ-SAC outperformed SAC in most tasks, achieving a higher success rate in fewer steps and significantly improving asymptotic performance and sample efficiency. The proposed method also demonstrated effectiveness in manipulation tasks.

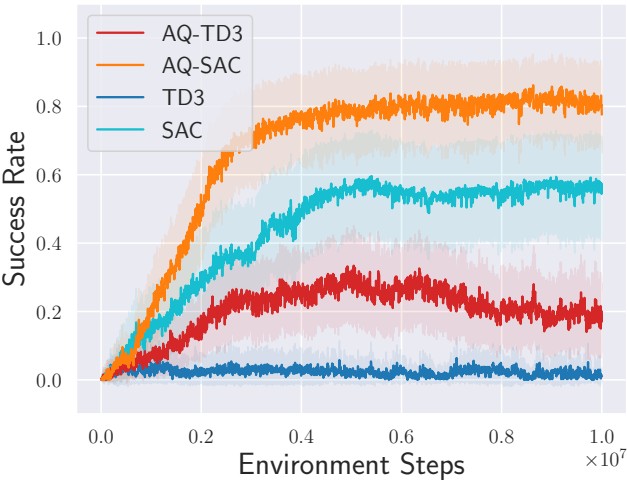

*Figure 5.* The average success rate across the 10 manipulation tasks in Meta-World.

### 4.3. Effects of $\tau$ Annealing

We evaluated the effectiveness of annealing $\tau$ using AQ-SAC. Table 2 presents the final average scores across all tasks for AQ-SAC when $\tau$ is annealed compared to when $\tau$ is fixed. AQ-SAC with $\tau$ annealing, starting from 0.9, outperformed the versions with fixed $\tau$ at 0.6, 0.7, 0.8, 0.9, and 0.95, highlighting the effectiveness of annealing. Figure 7 illustrates the results for two challenging tasks: humanoid-run and hopper-hop. In hopper-hop, when $\tau$ is fixed, increasing $\tau$ from 0.5 (SAC) to 0.6 and 0.7 improves the score, but the learning process becomes unstable, leading to a decline in scores at $\tau = 0.8$, with $\tau = 0.7$ producing the best result. In contrast, AQ-SAC, which anneals $\tau$ from 0.9, maintained stable learning and achieved higher final scores than the fixed-$\tau$ versions, even with the optimal fixed $\tau$ value of 0.7. In humanoid-run, the fixed $\tau = 0.7$ showed better early-stage learning than AQ-SAC, but the learning became unstable over time, resulting in a significant score drop by the end of the training. For larger $\tau$ values, the scores were considerably lower, suggesting high sensitivity to $\tau$ when it is not annealed. AQ-SAC, on the other hand, maintained relatively stable score improvement. The results for other tasks are provided in Appendix C.

Even when annealing $\tau$, the initial value of $\tau$ remains a hyperparameter. However, compared to fixed $\tau$, annealing makes the method less sensitive to this hyperparameter. As shown in Table 2, in the fixed $\tau$ case, $\tau = 0.7$ performed best, but increasing it to 0.8 led to a significant drop in scores, with even lower scores for $\tau = 0.9$ and 0.95. In the annealed $\tau$ case, the best initial value was $\tau_{\text{init}} = 0.9$. However, scores did not change significantly even when the initial value varied between 0.7 and 0.95, demonstrating that AQ-SAC is more robust to hyperparameter variations than the fixed $\tau$ case.

*Table 2.* The average final score across 10 DM Control tasks when $\tau$ is annealed compared to when $\tau$ is fixed in AQ-SAC. Annealing $\tau$ improves the scores and enhances robustness to $\tau$ settings.

| Method | Mean | IQM |
|---|---|---|
| Annealed (0.7) | 720.2 (694.8 - 741.0) | 821.7 (803.4 - 836.0) |
| Annealed (0.8) | 742.1 (729.1 - 754.7) | 824.0 (805.4 - 840.2) |
| Annealed (0.9) | **746.1** (732.0 - 758.5) | **832.4** (815.3 - 844.8) |
| Annealed (0.95) | 736.1 (725.7 - 745.9) | 815.6 (798.8 - 827.1) |
| Fixed (0.6) | 713.6 (690.5 - 734.5) | 822.7 (800.1 - 834.9) |
| Fixed (0.7) | 730.7 (715.3 - 745.7) | 825.0 (806.1 - 840.0) |
| Fixed (0.8) | 683.4 (660.8 - 702.9) | 775.9 (756.5 - 790.4) |
| Fixed (0.9) | 588.2 (559.0 - 613.6) | 632.7 (594.9 - 665.9) |
| Fixed (0.95) | 364.9 (338.8 - 390.6) | 303.5 (265.4 - 341.5) |

We measured the bias of the estimated Q-value with respect to the Monte Carlo return, and the results are presented in Figure 12. The bias increases as $\tau$ becomes larger, and in AQ-SAC, it eventually reaches a level comparable to that of SAC. This outcome is consistent with the preliminary experiments shown in Figure 2.

In AQ-L, a linear annealing schedule was used. We also conducted experiments with several non-linear annealing patterns. The results demonstrated that linear annealing achieves sufficiently good performance. The details of the experiments on non-linear annealing patterns are provided in Appendix H.

## 5. Related Work

**Off-Policy Online RL** In continuous action space online RL tasks, methods such as SAC (Haarnoja et al., 2018) and TD3 (Fujimoto et al., 2018) are widely used. SAC is an actor-critic method based on Haarnoja et al. (2017) that incorporates entropy maximization. The value function adds the policy's entropy to the target value and models the soft value function for the current policy. TD3 is an extension of DDPG (Lillicrap, 2015), incorporating techniques like clipped double Q-learning, action noise, and delayed policy updates. Like SAC, TD3 updates the value function based on SARSA, learning the value function under the current policy. Both methods employ SARSA-based updates, focusing on learning the value function for the current policy. Our study demonstrates that by replacing the value function update with the expectile loss, shifting to a Q-learning-based maximization, we can achieve improved performance.

Overestimation bias is a recurring issue in RL. Thrun & Schwartz (1993) formalized how noise in function approximators leads to overestimation bias, showing how noise variance and the number of actions contribute to the increase in bias. Lan et al. (2020) and Chen et al. (2021) suggest that overestimation can sometimes be beneficial for exploration while using ensembles to control the mean and variance of

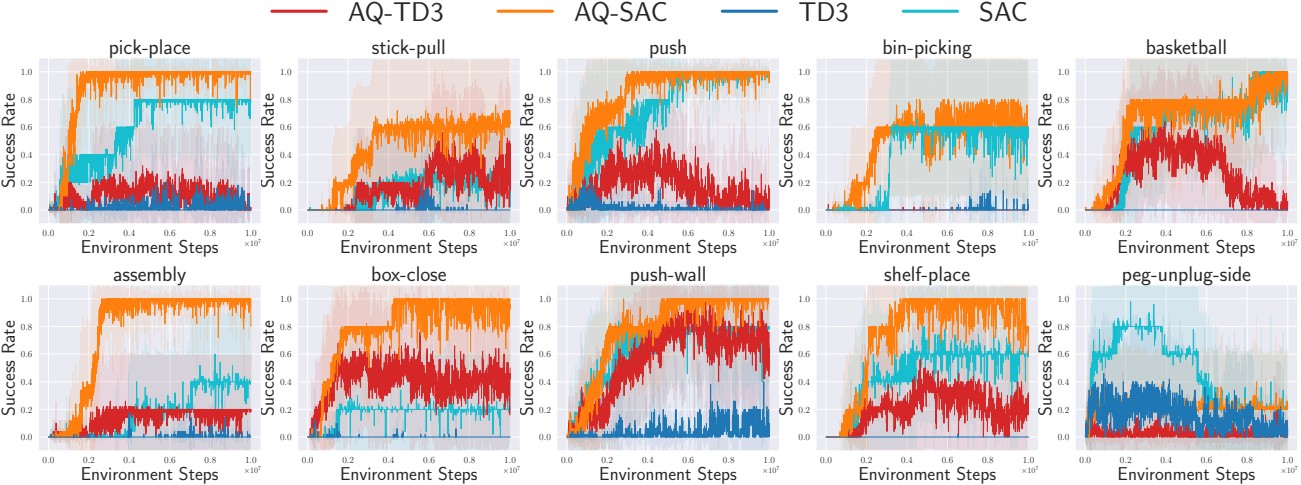

*Figure 6.* The average success rate for each task in Meta-World.

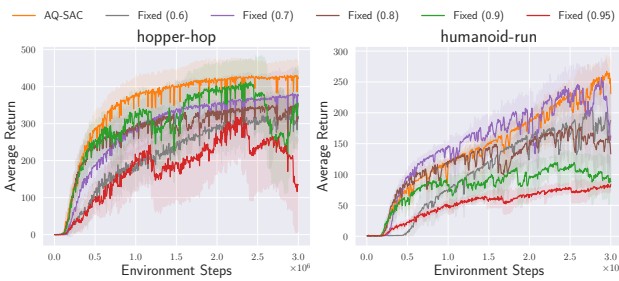

*Figure 7.* The average return of AQ-SAC on hopper-hop and humanoid-run when $\tau$ is annealed compared to when $\tau$ is fixed. Annealing $\tau$ improves asymptotic performance and stability.

bias. Incorporating these ensemble techniques into our proposed method could potentially enhance sample efficiency, which could be a promising direction for future work.

**Value Function Maximization in Continuous Action Space** In continuous action tasks, calculating the maximum Q-value is generally intractable. Various methods have tackled this issue by using asymmetric loss functions (Kostrikov et al., 2022; Garg et al., 2023; Xu et al., 2023; Omura et al., 2024; Sikchi et al., 2024), evaluating multiple actions (Kalashnikov et al., 2018; Kumar et al., 2019; 2020), or discretizing actions (Tavakoli et al., 2018; Seyde et al., 2023; 2024).

IQL (Kostrikov et al., 2022) approaches Q-value as a random variable with inherent randomness related to the actions, modeling the maximum value using the expectile loss with an expectile parameter close to 1. Adjusting the hyperparameter $\tau$ from 0.5 to 1 shifts the value function estimate from SARSA-based to Q-learning-based. However, it should be noted that IQL focuses on offline RL and

online fine-tuning following offline RL, and it is not an algorithm designed for learning without an offline dataset. XQL (Garg et al., 2023) models the soft optimal value in maximum entropy RL using Gumbel loss, derived from maximum likelihood estimation under the assumption of a Gumbel error distribution. MXQL (Omura et al., 2024) stabilizes the Gumbel loss by employing a Maclaurin expansion. By increasing the order of the expansion from 2 to infinity, MXQL transitions from SARSA-based learning to soft Q-learning-based learning. Although MXQL may benefit from annealing the expansion order to improve performance, tuning hyperparameters can be challenging due to the infinite range of the optimality parameter. The experiment combining MXQL and AQ-L is described in Appendix J.

Kalashnikov et al. (2018); Kumar et al. (2019; 2020) improve sample efficiency by modeling the optimal Q-value through sampling multiple actions from the policy distribution and selecting the action that maximizes the Q-value. These methods can shift from SARSA-based to Q-learning-based by increasing the number of samples from 1 to infinity. However, tuning poses challenges, and computational costs rise significantly as the number of samples increases.

Some methods, such as Tavakoli et al. (2018); Seyde et al. (2023; 2024); Ireland & Montana (2024), achieve optimal Q-value estimation through action discretization. However, if the number of intervals is small, accurate action selection becomes difficult while increasing the intervals reduces sample efficiency.

The most closely related prior work is Ji et al. (2024). Motivated by the underestimation of the value function in actor-critic methods, they compute a weighted sum of target values from the Bellman optimality operator, where the max

over actions is computed within the replay buffer via in-sample maximization, and from the Bellman operator. In contrast, our study aims to accelerate early-stage learning by introducing a beneficial bias, and to reduce this bias in the later stages. Rather than combining the outputs of the Bellman optimality operator and the Bellman operator, we propose a gradual transition from the Bellman optimality operator, where the max is computed under the current policy, to the Bellman operator over the course of training. While Ji et al. (2024) estimate two separate Q-values, our approach allows us to maintain a single Q-function throughout training. Therefore, our method can be viewed as a simplification of Ji et al. (2024), incorporating a novel motivational rationale and a scheduling mechanism to gradually reduce optimality, while relying on a single Q-function.

## 6. Conclusion

This study investigated the challenges and potential of modeling optimal value in online RL, particularly within actor-critic methods for continuous action tasks. Through experimental validation, we demonstrated that using the Bellman optimality operator accelerates learning but introduces overestimation bias, while the Bellman operator ensures stable value estimation at the cost of slower learning. To address this trade-off, we proposed Annealed Q-learning, which smoothly transitions from the Bellman optimality operator to the Bellman operator using the expectile loss. This approach initially enhances learning speed while mitigating bias in the later stages. Experiments on diverse locomotion and manipulation tasks validated the effectiveness of the proposed method, achieving superior performance compared to widely used algorithms like SAC and TD3, and demonstrating robustness to hyperparameter settings through the annealing process. While linear annealing contributed to performance improvement, further optimization of the annealing schedule is a promising direction for future research.

## Impact Statement

This paper presents work whose goal is to advance the field of Machine Learning. There are many potential societal consequences of our work, none which we feel must be specifically highlighted here.

## Acknowledgments

This research is partially supported by JST Moonshot R&D Grant Number JPMJPS2011, CREST Grant Number JP-MJCR2015 and Basic Research Grant (Super AI) of Institute for AI and Beyond of the University of Tokyo. T.O was supported by JSPS KAKENHI Grant Number JP25K03176. K.O. was supported by JST SPRING, Grant Number JP-MJSP2108.

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

## A. Details of Preliminary Experiments

In the preliminary experiment, we compared Q-learning-based updates and SARSA-based updates in actor-critic models using the environment shown in Figure 1. The details of this experiment are described below. The critic manages the estimated Q-values for each state-action pair in a table and updates them based on either Q-learning or SARSA. The actor manages logits for each state-action pair in a table, calculates the distribution using the softmax function, and samples actions from this distribution. Thus, the policy is expressed as follows:

$$\pi_\theta(a \mid s) = \frac{\exp(\theta_{s,a})}{\sum_b \exp(\theta_{s,b})}, \tag{8}$$

The update of these logits is performed using the policy gradient method, with the update equation given as follows:

$$\theta \leftarrow \theta + \alpha \nabla_\theta \log \pi_\theta(a_t \mid s_t) Q(s_t, a_t),$$
$$\nabla_{\theta_{s,a'}} \log \pi_\theta(a \mid s) = \delta_{a,a'} - \pi_\theta(a' \mid s), \tag{9}$$

where $\delta_{a,a'}$ is the Kronecker delta. The step size used for updates in both the critic and the actor was set to 1e-3. While increasing the step size accelerates learning, the observation that Q-learning-based updates are faster than SARSA-based updates remained consistent. A step size that yielded smooth learning curves was chosen. When using $s_0$ as the only initial state, the lower-value state $s_2$ was visited less frequently than $s_1$, making it challenging to accurately estimate the Q-value for $s_2$. To address this, the initial state was randomly selected from $s_0$, $s_1$, and $s_2$. Additionally, although actions were sampled from the policy, the estimation of Q-values for low-value actions progressed slowly. To mitigate this, a 10% probability of taking random actions, akin to an $\epsilon$-greedy policy, was introduced.

In Figure 2, the estimated Q-values for $s_0$ are shown, while the Q-values for $s_1$ and $s_2$ are presented in Figure 8. These results are from experiments where noise was added to the Q-values. However, since the next state for $s_1$ and $s_2$ is a terminal state and Q-values are learned solely from the reward, they are unaffected by the noise. Furthermore, the updates are identical for both Q-learning-based and SARSA-based methods under these conditions, leading to the same Q-value estimation in both cases.

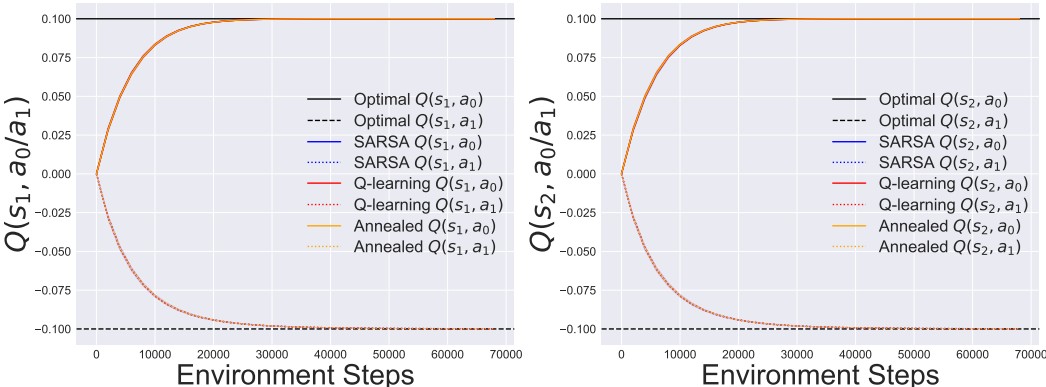

*Figure 8.* The estimated Q-values for $s_1$ and $s_2$ in the experiment where noise was added to reproduce the randomness of Q-value estimation in the environment of Figure 1.

### A.1. Results under Different Settings

**Variance of Noise**   As the standard deviation $\sigma$ of Gaussian noise $\mathcal{N}(0, \sigma)$ increases, the randomness of the estimated Q-value is expected to increase, leading to a larger overestimation bias. Therefore, experiments were conducted using various values of $\sigma$. The MDP is the same as in Figure 1 The results, shown in Figure 9, indicate that the bias increases as $\sigma$ becomes larger. Even with Annealed, a larger $\sigma$ causes bias in the early stages of learning; however, this promotes exploration, and ultimately, the values converge to those obtained with SARSA.

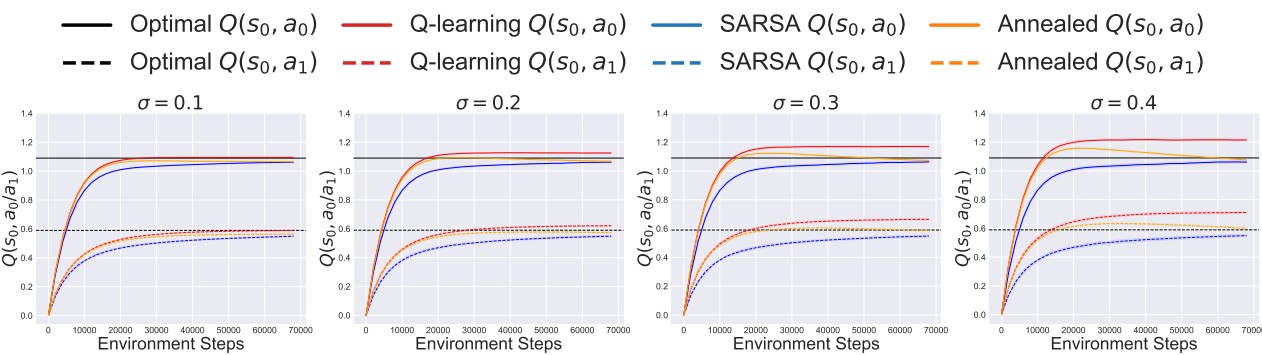

*Figure 9.* The estimated Q-values obtained using various standard deviations $\sigma$ of Gaussian noise in the environment of Figure 1.

$r_3$ **and** $r_4$    If the Q-values $Q(s', a')$ vary significantly depending on the action, then even if the Q-values are noisy, the maximum action Q-value $\max_{a'} Q(s', a')$ can be reliably selected each time, making overestimation less likely. Conversely, if $Q(s', a')$ does not change much across different actions $a'$, overestimation becomes more significant. To verify this, we varied $r_3$ and $r_4$ and estimated the Q-values at $s_0$. The standard deviation of the noise is 0.3. The results are shown in Figure 10. It was observed that when the difference between $r_3$ and $r_4$ is large, the bias decreases.

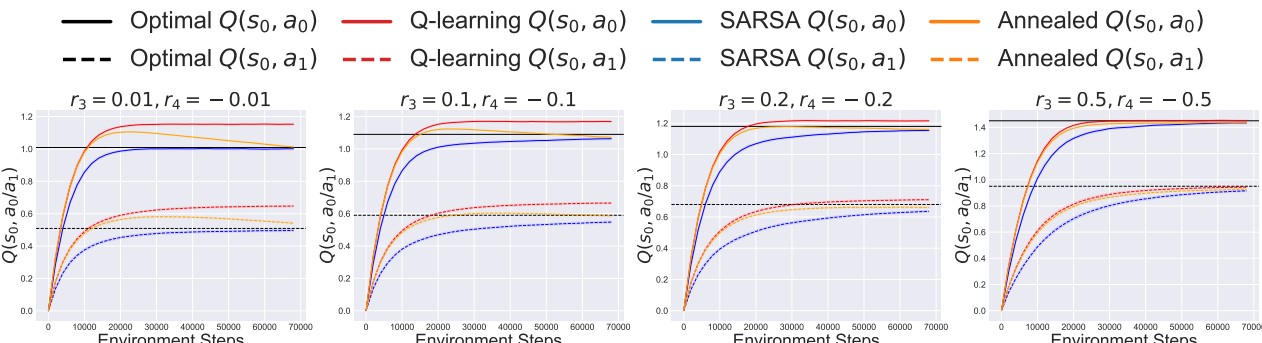

*Figure 10.* The estimated Q-values obtained using various values of $r_3$ and $r_4$ with the standard deviations $\sigma$ of Gaussian noise is 0.3 in the environment of Figure 1.

$r_1$ **and** $r_2$    In the preliminary experiment conducted in the environment shown in Figure 1, the Q-values at $s_0$ were both positive. We conduct experiments under conditions where the Q-value is negative. In Figure 1, the reward values were set as $r_1 = 1$ and $r_2 = 0.5$. While keeping the interval unchanged, we modify them to $r_1 = 0.25$ and $r_2 = -0.25$. The results obtained by varying the variance of the noise are shown in Figure 11. The results were generally similar to those in the case of positive rewards. However, due to the initial Q-value being set to zero, actions corresponding to negative Q-values were less likely to be selected, resulting in slower learning. Therefore, we extended the training steps. Despite this adjustment, SARSA, which inherently learns more slowly, failed to converge.

## B. Details of Experiments on DM Control and Meta-World

The implementations of TD3, SAC, AQ-TD3, and AQ-SAC are based on D'Oro et al. (2023). To ensure a fair comparison, all methods employed a batch size of 256, and both the actor and critic networks used two hidden layers consisting of 256 units each. XQL uses the official implementation, and we used XSAC, which integrates XQL with SAC. As described in the XQL paper, the temperature parameter $\beta$ was evaluated for values [1, 2, 5], and the best value $\beta = 5$ was chosen. Other hyperparameters for all methods follow the values reported in their respective papers. For tasks in the DM Control, the results are averaged over 10 random seeds, while, due to computational considerations, 5 random seeds are used to report the results for Meta-World tasks. The annealing period $T$ corresponds to the total number of training steps. Specifically, it is set to 3 million steps for DM Control tasks and 10 million steps for Meta-World tasks. The initial expetile value for

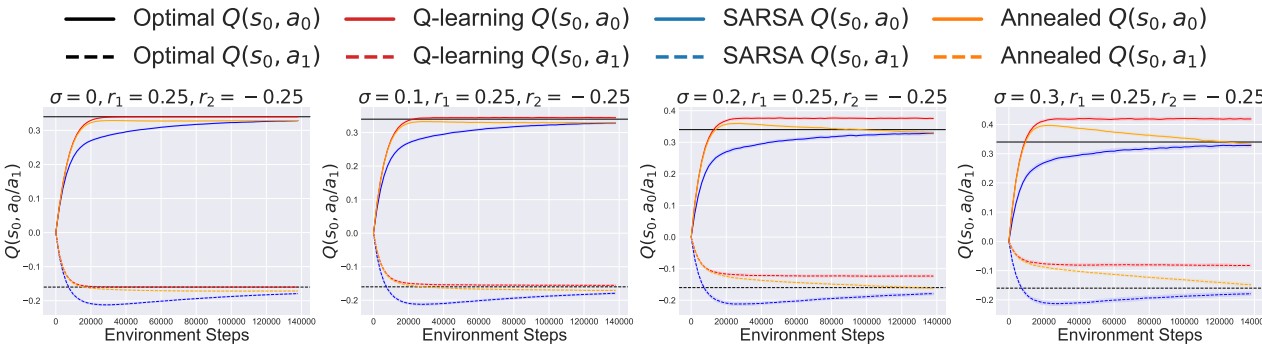

*Figure 11.* The estimated Q-values obtained using $r_1 = 0.25$ and $r_2 = -0.25$ with the various standard deviations $\sigma$ of Gaussian noise in the environment of Figure 1.

annealing, $\tau_{\text{init}}$, is 0.8 for AQ-TD3 and 0.9 for AQ-SAC in the DM Control tasks, and 0.7 for both AQ-TD3 and AQ-SAC in the Meta-World tasks. The hyperparameters of AQ-TD3 and AQ-SAC are summarized in Table 3.

We measured the bias of the estimated Q-value using AQ-SAC. Following the methodology of Chen et al. (2021), we evaluated the bias with respect to the Monte Carlo return. The results are presented in Figure 12. The bias increases as $\tau$ becomes larger, and in AQ-SAC, it eventually reaches a level comparable to that of SAC.

*Table 3.* Hyperparameters for AQ-TD3 and AQ-SAC.

| Parameter | AQ-TD3 | AQ-SAC |
|---|---|---|
| Discount factor | 0.99 | |
| Minibatch size | 256 | |
| Optimizer | Adam | |
| Learning rate | 0.0003 | |
| Activation function | ReLU | |
| Number of hidden layers | 2 | |
| Hidden units per layer | 256 | |
| Replay buffer size | $10^6$ | |
| $\tau$ (EMA coefficient for target networks) | 0.995 | |
| $\tau_{\text{init}}$ (DM Control) | 0.8 | 0.9 |
| $\tau_{\text{init}}$ (Meta-World) | 0.7 | |

*Table 4.* The average score at 1M steps across the DM Control tasks.

| Method | Mean | IQM |
|---|---|---|
| AQ-TD3 | **620.8** (609.9 - 631.5) | **655.4** (637.0 - 673.9) |
| AQ-SAC | **624.8** (611.4 - 637.5) | **651.5** (635.0 - 669.6) |
| TD3 | 376.8 (348.1 - 407.3) | 327.0 (281.5 - 377.1) |
| SAC | 493.7 (454.6 - 531.9) | 516.1 (451.7 - 578.5) |
| XQL | 433.6 (394.8 - 472.1) | 433.5 (372.2 - 492.9) |

## C. Experimental Results Using Fixed $\tau$ and Annealed $\tau$

To evaluate the effect of annealing, we compared the results of annealing $\tau$ and fixing $\tau$ using AQ-SAC. Figure 13 shows the average return for each task when annealing from various $\tau_{\text{init}}$ values. Figure 14 shows the average return for each task when using various fixed $\tau$ values. When annealing, the performance is less sensitive to the $\tau$ value, demonstrating increased robustness to hyperparameters through annealing.

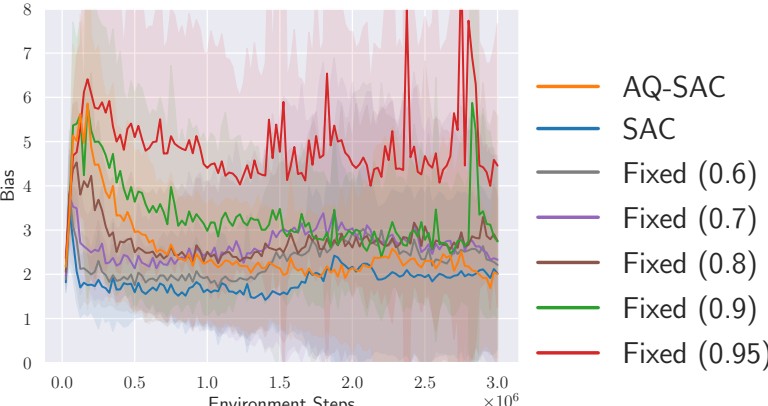

*Figure 12.* The bias of the estimated Q-value with respect to the Monte Carlo return. The larger $\tau$, the greater the bias, and in AQ-SAC, the bias eventually became comparable to that of SAC.

## D. Experimental Results of Max-backup

In continuous action tasks, one straightforward way to compute maximum Q-values is by sampling multiple actions from the current policy, calculating their Q-values, and selecting the maximum. This method, known as max-backup, was employed by Kumar et al. (2020). In this study, we compared max-backup with AQ-SAC using different sampling counts. Both methods were implemented based on SAC. Figure 15 presents the average return for each task when using max-backup with various action sampling numbers. Max-backup is implemented based on Kumar et al. (2020). While max-backup has the drawback of increased computational cost, it demonstrated scores comparable to AQ-SAC in some tasks. However, in more challenging tasks such as hopper-hop and humanoid-run, AQ-SAC outperformed max-backup. This suggests that expectile-based methods for maximum value estimation are more effective than the sampling-based approach used in max-backup. Since the computation cost of max-backup increases with the number of samples, AQ-SAC is also superior in terms of both performance and computational efficiency.

## E. Analysis of Annealing Duration

In AQ-L, annealing is performed over the entire training steps, but the annealing duration can also be a hyperparameter. Therefore, experiments were conducted using AQ-SAC with varying annealing durations. The results are shown in Figure 16. After the annealing phase, learning proceeds with $\tau = 0.5$, consistent with the standard SAC setting. Notably, even when $T$ is as small as 1 million steps, there is a significant performance improvement compared to SAC. The final return does not vary significantly across different values of $T$, indicating that AQ-SAC is robust to $T$ within the range tested.

## F. Results from Extended Training on *humanoid-run* and *humanoid-walk*

In Figure 4, both humanoid-run and humanoid-walk had not yet converged even after 3 million steps, and their performance continued to improve. Therefore, the results of extending training up to 10 million steps for both AQ-SAC and SAC are shown in Figure 17. Even in terms of the asymptotic performance after 10 million steps of training, the proposed method AQ-SAC outperformed SAC.

## G. The Necessity of the $V$-function

In IQL, both the Q-function and V-function were trained to handle environmental stochasticity. However, as shown in Figure 18, this study suggests that training with only the Q-function achieves better performance even in stochastic environments. Therefore, AQ-L utilizes only the Q-function.

Figure 18 presents results comparing AQ-SAC using only the Q-function versus AQ-SAC using both the Q-function and V-function in the stochastic hopper-hop environment. Stochasticity in the environment is introduced by adding Gaussian noise with a mean of 0 to the actions input into the DM Control tasks. The left figure shows the results when $\tau$ is annealed,

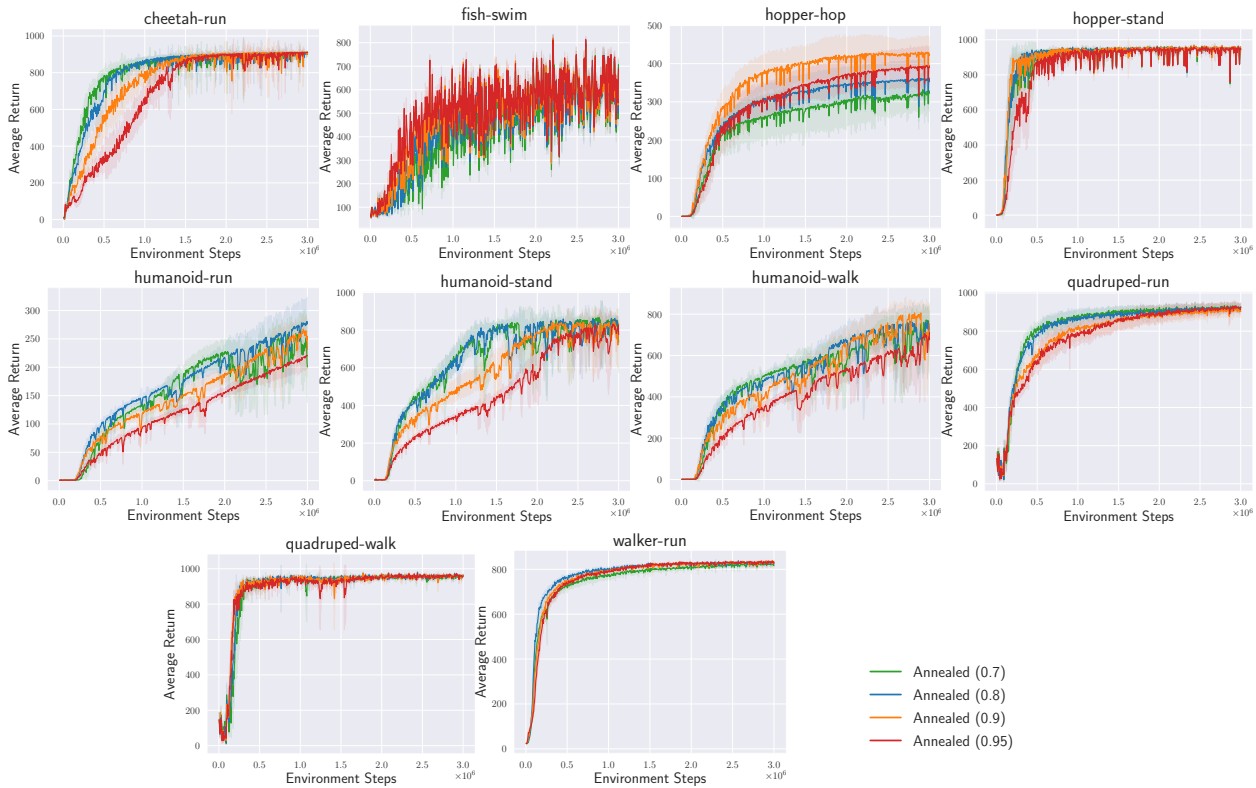

*Figure 13.* The average return for each task when annealing with various $\tau_{\text{init}}$ values in AQ-SAC.

while the right figure corresponds to fixed $\tau$. The values used were $\tau_{\text{init}} = 0.9$ for annealing and $\tau = 0.7$ for the fixed case, as these settings yielded the best performance.

In both cases, whether annealing or fixed, using only the Q-function consistently outperformed using both the Q-function and V-function across all levels of noise standard deviation. This performance difference is attributed to the additional network required when using the V-function, which increase approximation error and degrade performance.

For the annealing case in the left figure, the degradation in average return as noise standard deviation increases is similar whether or not the V-function is used. This is likely because annealing results in learning behavior similar to SAC toward the end of training, mitigating the impact of environmental randomness on the IQL loss.

In the fixed $\tau$ case shown in the right figure, performance degradation due to increased noise standard deviation is more pronounced when the V-function is not used. Nevertheless, using only the Q-function still outperforms the setup with both Q-function and V-function.

## H. Non-linear Annealing Patterns

We also tested non-linear annealing patterns with AQ-SAC. Figure 19 illustrates different annealing patterns, including exponential annealing (Exp1) as proposed by Morerio et al. (2017) in the context of dropout scheduling, its inverse (Exp2), and sigmoid-based annealing (Sigmoid). The results, as shown in Table 5, indicate that Sigmoid and Exp1 performed similarly to linear annealing, but Exp2 resulted in worse performance. This suggests that prolonging the high $\tau$ period leads to excessive initial bias, which negatively impacts the later stages of learning. These observations indicate that linear annealing proves to be effective enough, and future research could explore dynamic adjustments based on bias.

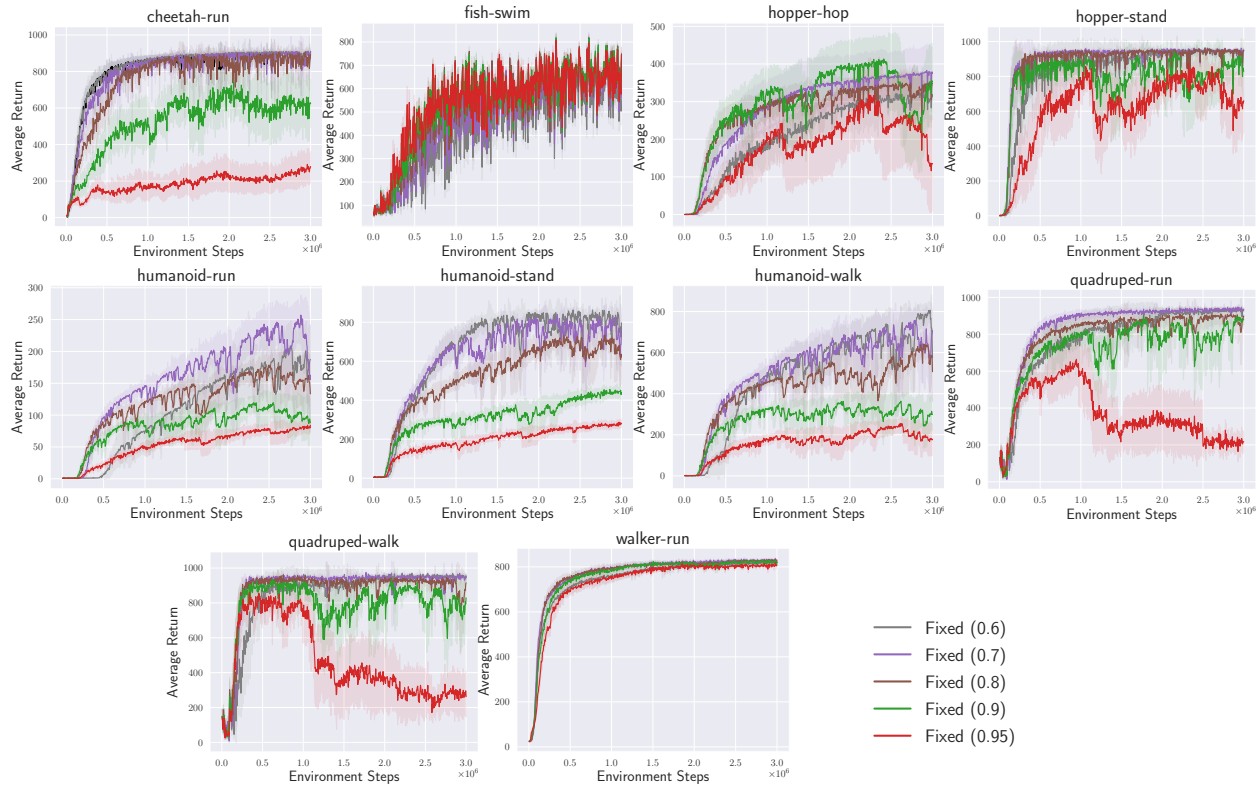

*Figure 14.* The average return for each task when using various fixed $\tau$ values in AQ-SAC.

*Table 5.* The average scores across the 10 DM Control tasks for AQ-SAC using various annealing patterns. A simple linear annealing showed the best performance.

| Method | Mean | IQM |
|---|---|---|
| Linear | 746.1 (732.0 - 758.7) | 832.4 (815.1 - 844.8) |
| Exp1 | 728.1 (716.7 - 739.4) | 809.1 (792.9 - 823.7) |
| Exp2 | 694.9 (680.5 - 706.8) | 772.4 (759.3 - 784.2) |
| Sigmoid | 742.2 (731.3 - 752.9) | 812.9 (795.7 - 829.4) |

## I. The Relationship Between Bias and Exploration

We measured policy entropy during early training. The table below shows average entropy over DMC tasks for SAC (Fixed (0.5), (0.7), and (0.9)), corresponding to SAC with expectile loss with $\tau = 0.5, 0.7, 0.9$. As shown in Figure 12, these methods exhibit overestimation bias in the order: Fixed (0.9) > Fixed (0.7) > SAC. The entropy follows the same trend, suggesting higher bias leads to broader exploration.

This supports the intuitive hypothesis that overestimation can lead to suboptimal actions due to inflated Q-values, promoting exploration. Prior work (e.g., Section 3 of Lan et al. (2020)) also demonstrates that overestimation can be beneficial in tasks where exploration is important, while it degrades performance in tasks where exploration is undesirable. These results support the claim that overestimation bias promotes exploration.

## J. Combination of Annealed Q-learning and MXQL

In our study, we enable the transition from the Bellman optimality operator to the Bellman operator by using expectile regression from IQL (Kostrikov et al., 2022). Gumbel regression, proposed in XQL (Garg et al., 2023), allows the computation of the soft Bellman optimality operator. In MXQL (Omura et al., 2024), which applies the Maclaurin expansion

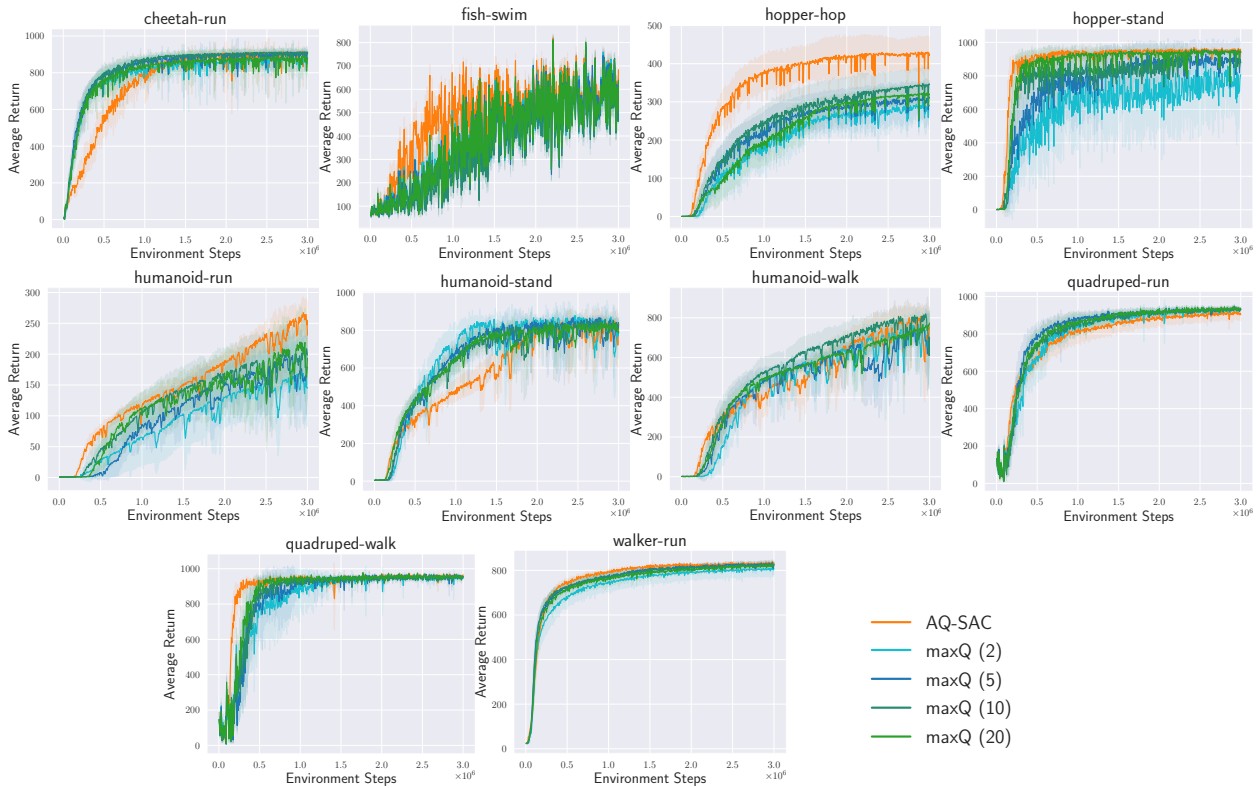

*Figure 15.* The average return for each task when using max-backup with various action sampling numbers.

*Table 6.* The average policy entropy in 10 DM Control tasks. 15k steps corresponds to shortly after training begins at 10k steps. These results suggest that a large $\tau$ in the early stages of training increases policy entropy and promotes exploration.

| Method | 15k steps | 100k steps | 200k steps |
|---|---|---|---|
| Fixed (0.5) (SAC) | $6.78 \pm 0.30$ | $6.23 \pm 0.39$ | $5.97 \pm 0.38$ |
| Fixed (0.7) | $7.41 \pm 0.36$ | $6.40 \pm 0.40$ | $6.11 \pm 0.38$ |
| Fixed (0.9) | $8.28 \pm 0.44$ | $6.94 \pm 0.39$ | $6.52 \pm 0.38$ |

to XQL, it is possible to transition from the soft Bellman optimality operator to the Bellman operator by varying the order of the expansion. Therefore, in this section, we conduct experiments by combining MXQL and AQ-L. The expansion order of MXQL was linearly decayed from $n_{\text{init}}$ to 2 in steps of 2. Experiments were conducted in combination with TD3 and SAC. We tested $n_{\text{init}} \in 4, 8, 12$ and adopted 4, which yielded the best average score. The hyperparameter $\beta$ of MXQL was tested over $0.1, 0.5, 1, 2, 5$, and we adopted 1, which resulted in the best average score. The overall average score in DM Control is shown in Figure 20, and the scores for individual tasks are shown in Figure 21. Even when $n_{\text{init}} = 4$, the computation includes fourth-order terms, the loss function changes discretely, and the training is sensitive to $\beta$, which makes the learning unstable. Nevertheless, the performance is significantly improved when combined with TD3.

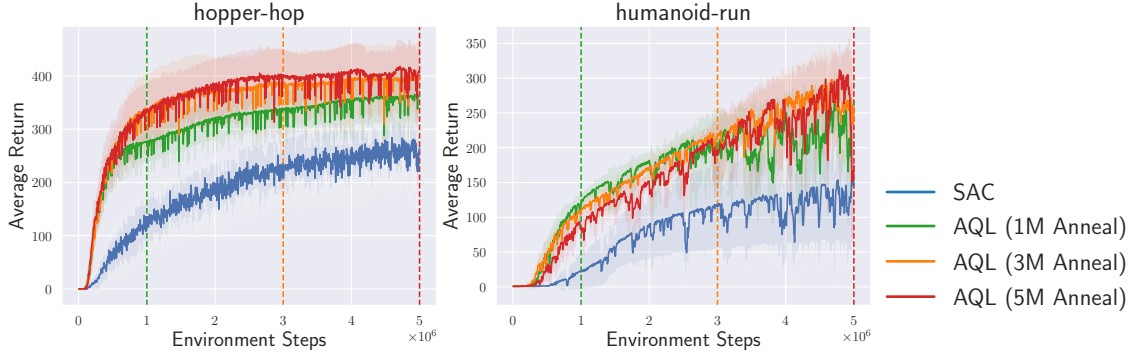

*Figure 16.* The average return of AQ-SAC when the annealing duration is varied. The dashed line represents the step count at which annealing ends, after which learning proceeds with $\tau = 0.5$, the same as the SAC.

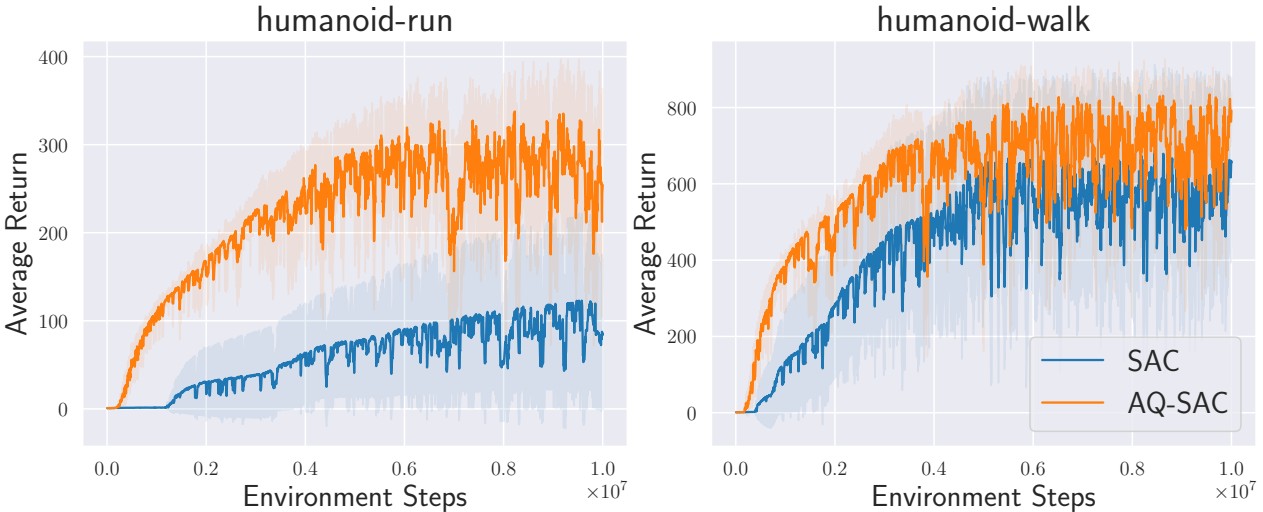

*Figure 17.* The averate return of SAC and AQ-SAC. In AQ-SAC, training continues with $\tau = 0.5$ after 3M steps of annealing.

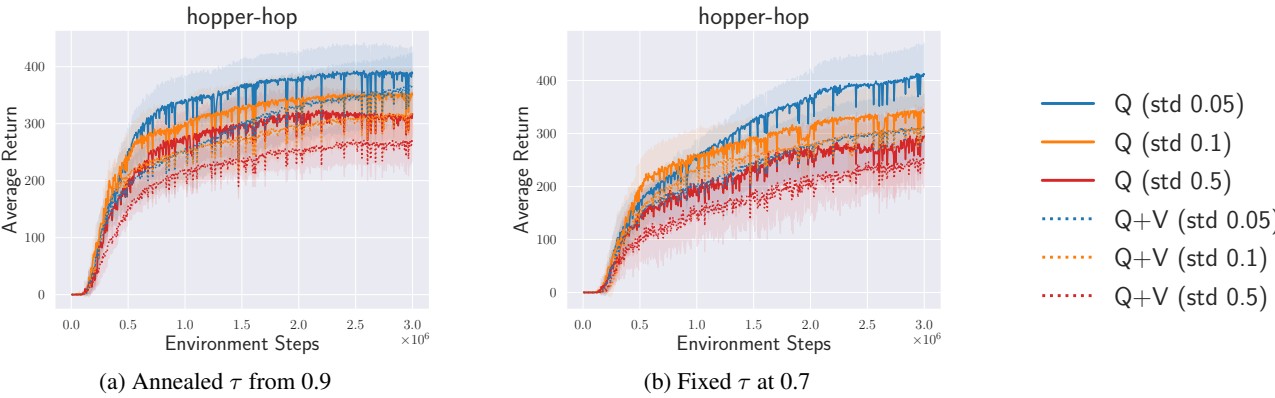

(a) Annealed $\tau$ from 0.9          (b) Fixed $\tau$ at 0.7

*Figure 18.* The average scores for AQ-SAC trained using only the Q-function compared to AQ-SAC utilizing both the Q-function and V-function in a stochastic hopper-hop environment. The stochastic environment is created by adding zero-mean Gaussian noise to the actions fed into the DM Control environment. The different colors represent the varying standard deviations of the Gaussian noise applied. The solid lines represent results obtained using only the Q-function, while the dotted lines indicate those obtained using both the Q-function and V-function. The left figure shows the results when $\tau$ is annealed from an initial value of 0.9, and the right figure shows the results when $\tau$ is fixed at 0.7.

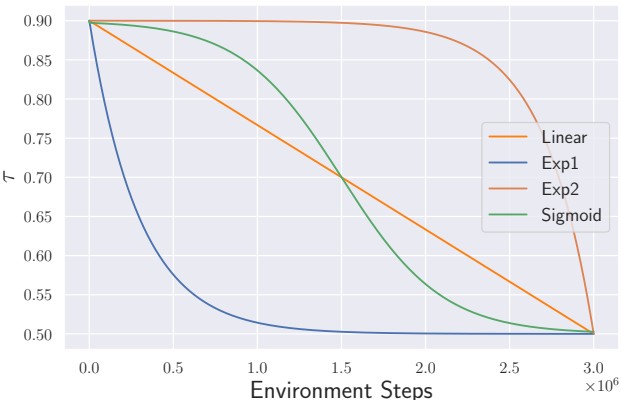

*Figure 19.* The annealing patterns of $\tau$ used in the experiments.

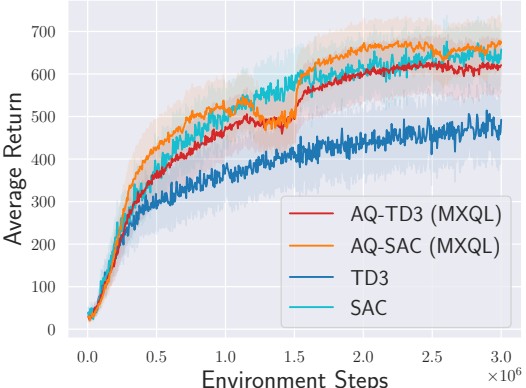

*Figure 20.* The average scores across the 10 locomotion tasks in DM Control.

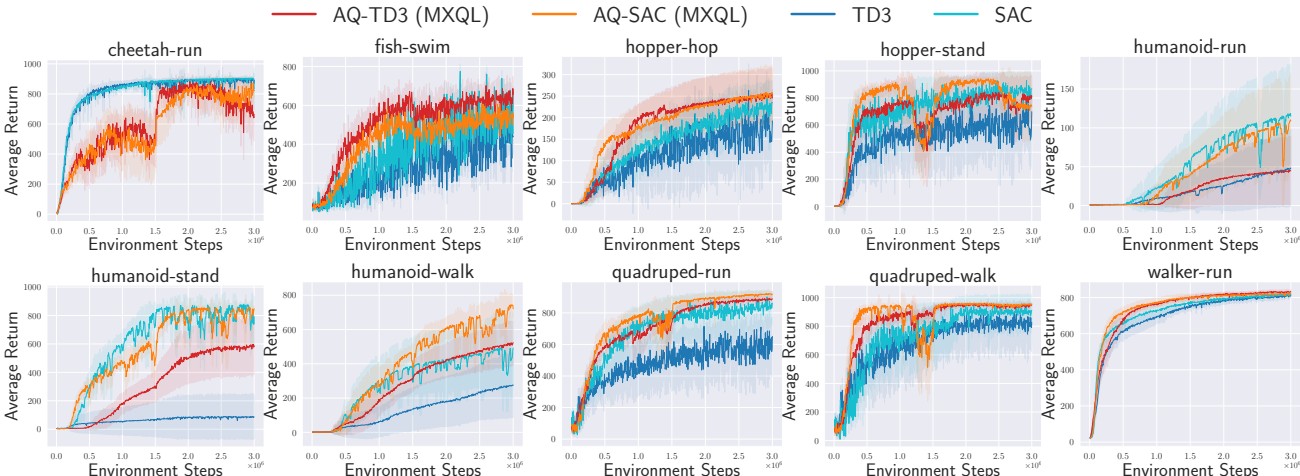

*Figure 21.* The average return for each task in DM Control.

