# OpenReview forum: "Gradual Transition from Bellman Optimality Operator to Bellman Operator in Online Reinforcement Learning"
_ICML.cc/2025/Conference — ICML 2025 poster_

### Official Review · Reviewer_v2xt · 2025-03-12

**Overall Recommendation:** 5

**Summary:**

The paper studies the contrast and connections between Bellman operator and Bellman optimality operator in online Reinforcement Learning (RL). Bellman operator is widely used for on-policy actor-critic methods but often hurts to sample efficiency. Bellman optimality operator, on the other hand, is widely used for discrete RL but suffers from overestimation bias in target values. The paper bridges the gap between the two operators by leveraging IQL's expectile value loss. Specifically, authors propose to gradually transition from the optimality operator to the traditional SARSA operator by annealing the expectile coefficient $\tau$. Contrary to IQL, updates only utilize the $Q$ value function in the target instead of the state-value function. Experiments on a small toy MDP, DM Control tasks and MetaWorld benchmark validate that annealing addresses overestimation and provides sample efficiency.

**Claims And Evidence:**

Please refer to strengths and weaknesses.

**Essential References Not Discussed:**

Please refer to strengths and weaknesses.

**Experimental Designs Or Analyses:**

Please refer to strengths and weaknesses.

**Methods And Evaluation Criteria:**

Please refer to strengths and weaknesses.

**Other Comments Or Suggestions:**

NA

**Other Strengths And Weaknesses:**

### Strengths

* This paper is exceptionally well written and easy to understand.
* The work investigates the annealing scheme thoroughly using comprehensive experiments.
* The paper tackles important problems of sample efficiency and overestimation bias by proposing a simple idea.
### Weaknesses

*  **Evaluation:** While the paper studies and analyzes the annealing scheme, its evaluation and overall comparative presentation could be improved. Authors could utilize the rliable library [1] or metrics to compare between different baselines. For instance, while authors compare the IQM, the optimality gap should also be compared and presented. Similarly, authors should also consider evaluating the gap between actual and estimated target $Q$ values to evaluate overestimation bias for atleast one of the DM Control tasks.
* **Contribution and Novelty**: While the paper makes a solid empirical contribution, its theoretical understanding remains an open question. Could the authors elaborate on how the annealing scheme affects the ability of IQL in extracting the optimal policy? Additionally, authors could discuss how the annealing scheme is different from prior efforts that aim to address overestimation bias and sample efficiency. In its current form the work only ablates and tunes $\tau$, a trick partially explored by IQL [2].

[1]. Agarwal et al, Deep Reinforcement Learning at the Edge of the Statistical Precipice, NeurIPS 2021.
[2]. Kostrikov et al, Offline Reinforcement Learning with Implicit Q-Learning, ICLR 2022.

**Questions For Authors:**

Please refer to strengths and weaknesses.

**Relation To Broader Scientific Literature:**

Please refer to strengths and weaknesses.

**Theoretical Claims:**

Please refer to strengths and weaknesses.

---

> ### Author Rebuttal · Authors · 2025-03-31
>
> Thank you very much for the highly constructive feedback. We provide several responses below.
>
> **Evaluation using optimality gap**
>
>  Thank you for the suggestion. As we understand it, the optimality gap refers to how much the agent's performance falls short of a target score, such as a human-level or oracle performance. In our tasks, however, such target scores are not always clearly defined. If the maximum achievable score in the task (e.g., 1000 in DM Control) is considered as the target, then we assume the optimality gap corresponds to the difference between 1000 and the agent’s average score. If this interpretation is incorrect, we would greatly appreciate clarification.
>
> **Evaluating overestimation via Q-value gap**
>
>  In Figure 12, we report the gap between the estimated Q-values and the actual Q-values computed via Monte Carlo return, across our proposed method and several ablations. In AQ-SAC, our proposed method, we observe that the bias is initially larger—potentially encouraging exploration—but gradually decays to a level comparable with the SAC (SARSA-based method) toward the end. This behavior aligns with our intended design.
>
>
> **Theoretical basis and impact of the annealing scheme**
>
> If we focus solely on optimality, when $\tau$ = 0.5, the max operator in Equation (3) of our paper is replaced by an expectation, turning the inequality into an equality. As a result, no bias is introduced, allowing for a more optimal value estimation. On the other hand, $\tau \simeq 1$ corresponds to the max operator, which introduces bias. Therefore, we anneal $\tau$ toward 0.5 during training to reduce bias and eventually enable the estimation of an optimal policy.
> It is an open question, as the reviewer correctly pointed out, how the max operator and bias affect the speed of improvement, and it is theoretically challenging to characterize this relationship. The table included in our response to Q2 of reviewer dvAE provides additional empirical evidence suggesting that bias can promote exploration, thereby highlighting the effectiveness of our method.
>
>
> Thank you again for the valuable feedback. If you have any further concerns, we would be happy to address them.

---

> > ### Comment · Reviewer_v2xt · 2025-04-02
> >
> > I thank the authors for their response to my comments. Below are my remain concerns-
> >
> > **Evaluation using optimality gap:** The authors' interpretation of the optimality gap is correct. Authors could measure this to compare between their ablations. Additionally, authors could simply utilize the rliable library or an equivalent framework for metrics. In its current form, the IQM comparison makes the evaluation less informative.

---

> > > ### Author Response · Authors · 2025-04-08
> > >
> > > Thank you once again for your valuable feedback. As suggested, we have evaluated IQM, Mean, and Optimality Gap using rliable. The results are presented below. We will incorporate these results in the camera-ready version of the paper. We sincerely appreciate the time and effort you dedicated to providing such insightful comments.
> > >
> > > | Algorithm | IQM (CI)                          | Mean (CI)                         | Optimality Gap (CI)             |
> > > |-----------|----------------------------------|-----------------------------------|---------------------------------|
> > > | AQ-TD3    | 820.000 (CI: 811.488 - 826.896)  | 740.304 (CI: 731.038 - 749.247)   | 259.696 (CI: 250.753 - 268.962) |
> > > | AQ-SAC    | 832.442 (CI: 815.114 - 844.640)  | 746.092 (CI: 732.023 - 758.475)   | 253.908 (CI: 241.525 - 267.977) |
> > > | TD3       | 516.018 (CI: 462.334 - 571.874)  | 492.670 (CI: 459.869 - 526.973)   | 507.330 (CI: 473.027 - 540.131) |
> > > | SAC       | 765.028 (CI: 712.900 - 800.936)  | 657.879 (CI: 623.902 - 689.004)   | 342.121 (CI: 310.996 - 376.098) |
> > > | XQL       | 628.837 (CI: 560.269 - 687.952)  | 564.443 (CI: 521.825 - 604.427)   | 435.557 (CI: 395.573 - 478.175) |

---

### Official Review · Reviewer_KfGB · 2025-03-14

**Overall Recommendation:** 3

**Summary:**

This paper is mainly concerned about improving the online policy learning. The paper is especially focusing on the respective advantages and disadvantages of Bellman (SARSA-like) and Bellman optimality (Q-learning-based) operators. Through training in a simple discrete action environment, they empirically showcase that Bellman optimality operator accelerates learning, while Bellman operator is less biased. Motivated by such balance, they use a linear combination of both target values from the two groups of methods to be their actual target in training. And to approximate taking maximum for Q-learning part, an expectile loss with linearly decayed threshold value is applied. The performance of such proposal is tested in several environments.

**Claims And Evidence:**

Although mainly using empirical evidence, the efficiency of the proposal is quite convincing.

**Essential References Not Discussed:**

None.

**Experimental Designs Or Analyses:**

- The motivating examples in Section 3 provides an intuition of the balance between two operators.
- Why in cheetah-run and humanoid-stand environments, the proposed methods are less efficient than some oracle methods? Especially it seems that SAC is still the best in the latter one?
- For the results of biases in Figure 12, how about annealed $\\tau$?
- Different annealing patterns are tested for the final choice of linear decay.

**Methods And Evaluation Criteria:**

- What is the justification for the linear decay in $w$?

**Other Comments Or Suggestions:**

None.

**Other Strengths And Weaknesses:**

> **Clarity**
- Line 212-213: what is the expectation over (where is the randomness)?

**Questions For Authors:**

None.

**Relation To Broader Scientific Literature:**

Such method provides a promising approach to balance between training efficiency and bias control.

**Theoretical Claims:**

N/A

---

> ### Author Rebuttal · Authors · 2025-03-31
>
> Thank you for the valuable feedback. Please find our responses below.
>
> **What is the justification for the linear decay in w?**
>
>  As shown in Table 4 and Figure 18, we experimented with several annealing patterns and found that linear decay achieved sufficiently good performance. While other patterns or adaptive strategies could also be viable design choices, we prioritized simplicity in this work.
>
> **Why in the cheetah-run and humanoid-stand environments, are the proposed methods less efficient than some oracle methods? Especially, it seems that SAC is still the best in the latter one?**
>
> In the cheetah-run environment, even the base algorithms such as SAC and TD3 converge relatively quickly to high performance. This suggests that sufficient exploration and improvement in policy optimality are already achieved without the need for further bias-induced acceleration. In such cases, the additional bias introduced by AQL may outweigh its benefits, leading to lower overall efficiency. This interpretation is supported by Figure 13, where reducing the initial value of $\tau$ helps mitigate the bias and results in convergence speeds comparable to SAC.
>
> A similar explanation applies to the humanoid-stand environment. Here, a high initial $\tau$ value of 0.9 likely introduces excessive bias, again offsetting the intended benefits of AQL. When $\tau$ is reduced to 0.7 or 0.8, the convergence becomes comparable to SAC, which further supports this hypothesis.
>
> These observations indicate that the inefficiency of our method in these environments can be mitigated through appropriate tuning of $\tau$. Importantly, regardless of the $\tau$ setting, the final returns achieved by our method remain competitive highlighting the robustness of the proposed approach.
>
> **For the results of biases in Figure 12, how about annealed $\tau$?**
>
>  The AQ-SAC results in Figure 12 correspond to the case where $\tau$ is annealed from 0.9 . In AQ-SAC, the initial bias increases, which can contribute to exploration, and then gradually decreases to a level comparable to the SARSA-based approach (SAC) toward the end. This behavior aligns with our intention. Similar trends were observed when annealing from different initial values of $\tau$, and we will include these additional results in the camera-ready version.
>
> **Line 212–213: what is the expectation over (where is the randomness)?**
>
>  The expectation is over the randomness in the Q-function, specifically the variable $\epsilon$ mentioned in the following sentence. We will add a clearer explanation to improve readability.
>
> We sincerely appreciate your helpful feedback once again. We hope that our responses have sufficiently addressed your concerns. If there are any remaining issues, we would greatly appreciate your guidance on what further explanations or revisions would be necessary to improve your score.

---

> > ### Comment · Reviewer_KfGB · 2025-04-07
> >
> > Thank you for the clarification! Please find my follow-up question below:
> >
> > > **Choice of $w$**
> > - Figure 18 seems to justify the choice of $\tau$ instead of $w$?

---

> > > ### Author Response · Authors · 2025-04-08
> > >
> > > Thank you for your follow-up question. As you correctly pointed out, Figure 18 illustrates the scheduling of $\tau$, not $w$ — we apologize for the confusion. To justify the use of a linear schedule for $w$ as well, We conducted preliminary experiments using the same scheduling approach as that in Figure 18, applied to $w$ as in Figure 2.
> > >
> > > In these experiments, we evaluated the estimation errors of Q(s₀, a₀) and Q(s₀, a₁) when using Gaussian noise with standard deviations of 0.2, 0.3, and 0.4 to the target Q-values, reporting the errors every 10,000 steps. The results are summarized in the tables below. While there are cases where non-linear scheduling may offer better performance, our findings suggest that a simple linear schedule works sufficiently well for $w$ in many cases.
> > >
> > > We will include clearer figures of these results in the camera-ready version. We hope these additional experiments help address your concerns and would greatly appreciate your consideration of a score update. Once again, thank you for your valuable feedback.
> > >
> > > ### $\sigma = 0.2$
> > >
> > > **Table for $a_0$**
> > >
> > > | Method | 0 | 10000 | 20000 | 30000 | 40000 | 50000 | 60000 |
> > > | --- | --- | --- | --- | --- | --- | --- | --- |
> > > | SARSA  | -1.090 | -0.225 | -0.080 | -0.055 | -0.044 | -0.038 | -0.031 |
> > > | Q-learning  | -1.090 | **-0.134** | 0.019 | 0.034 | 0.036 | 0.036 | 0.036 |
> > > | Linear  | **-1.090** | -0.146 | **-0.003** | **0.002** | **-0.006** | **-0.012** | **-0.018** |
> > > | Exp1  | -1.090 | -0.188 | -0.068 | -0.053 | -0.044 | -0.038 | -0.031 |
> > > | Exp2  | -1.090 | -0.134 | 0.019 | 0.035 | 0.034 | 0.033 | 0.025 |
> > > | Sigmoid  | -1.090 | -0.135 | 0.012 | 0.015 | -0.009 | -0.023 | -0.027 |
> > >
> > >
> > > **Table for $a_1$**
> > >
> > > | Method | 0 | 10000 | 20000 | 30000 | 40000 | 50000 | 60000 |
> > > | --- | --- | --- | --- | --- | --- | --- | --- |
> > > | SARSA  | **-0.590** | -0.209 | -0.124 | -0.090 | -0.070 | -0.058 | -0.048 |
> > > | Q-learning  | -0.590 | -0.133 | -0.030 | 0.004 | 0.019 | 0.026 | 0.031 |
> > > | Linear  | -0.590 | -0.137 | -0.045 | -0.019 | -0.013 | **-0.012** | **-0.013** |
> > > | Exp1  | -0.590 | -0.173 | -0.102 | -0.076 | -0.062 | -0.052 | -0.045 |
> > > | Exp2  | -0.590 | **-0.133** | **-0.030** | **0.004** | 0.018 | 0.025 | 0.025 |
> > > | Sigmoid  | -0.590 | -0.134 | -0.034 | -0.008 | **-0.007** | -0.014 | -0.019 |
> > >
> > > ### $\sigma = 0.3$
> > >
> > > **Table for $a_0$**
> > >
> > > | Method | 0 | 10000 | 20000 | 30000 | 40000 | 50000 | 60000 |
> > > | --- | --- | --- | --- | --- | --- | --- | --- |
> > > | SARSA  | -1.090 | -0.227 | -0.081 | -0.055 | -0.046 | -0.038 | -0.031 |
> > > | Q-learning  | -1.090 | -0.096 | 0.063 | 0.077 | 0.081 | 0.077 | 0.078 |
> > > | Linear  | **-1.090** | -0.108 | **0.030** | **0.028** | 0.016 | **0.001** | **-0.010** |
> > > | Exp1  | -1.090 | -0.172 | -0.064 | -0.051 | -0.042 | -0.038 | -0.030 |
> > > | Exp2  | -1.090 | **-0.096** | 0.062 | 0.077 | 0.077 | 0.075 | 0.062 |
> > > | Sigmoid  | -1.090 | -0.098 | 0.052 | 0.047 | **0.011** | -0.016 | -0.024 |
> > >
> > > **Table for $a_1$**
> > >
> > > | Method | 0 | 10000 | 20000 | 30000 | 40000 | 50000 | 60000 |
> > > | --- | --- | --- | --- | --- | --- | --- | --- |
> > > | SARSA  | -0.590 | -0.210 | -0.122 | -0.089 | -0.069 | -0.056 | -0.047 |
> > > | Q-learning  | -0.590 | -0.094 | 0.013 | 0.047 | 0.062 | 0.069 | 0.073 |
> > > | Linear  | -0.590 | -0.105 | -0.011 | **0.010** | **0.013** | 0.009 | **0.003** |
> > > | Exp1  | -0.590 | -0.156 | -0.091 | -0.070 | -0.059 | -0.050 | -0.043 |
> > > | Exp2  | -0.590 | **-0.093** | 0.012 | 0.046 | 0.059 | 0.066 | 0.066 |
> > > | Sigmoid  | **-0.590** | -0.096 | **0.007** | 0.030 | 0.024 | **0.007** | -0.006 |
> > >
> > > ### $\sigma = 0.4$
> > >
> > > **Table for $a_0$**
> > >
> > > | Method | 0 | 10000 | 20000 | 30000 | 40000 | 50000 | 60000 |
> > > | --- | --- | --- | --- | --- | --- | --- | --- |
> > > | SARSA  | -1.090 | -0.225 | -0.082 | -0.054 | -0.043 | -0.038 | -0.030 |
> > > | Q-learning  | -1.090 | -0.053 | 0.109 | 0.125 | 0.127 | 0.126 | 0.126 |
> > > | Linear  | **-1.090** | -0.071 | 0.065 | 0.058 | 0.038 | 0.019 | **-0.000** |
> > > | Exp1  | -1.090 | -0.155 | **-0.055** | **-0.049** | -0.044 | -0.037 | -0.031 |
> > > | Exp2  | -1.090 | **-0.053** | 0.109 | 0.126 | 0.124 | 0.119 | 0.102 |
> > > | Sigmoid  | -1.090 | -0.056 | 0.096 | 0.082 | **0.033** | **-0.008** | -0.023 |
> > >
> > > **Table for $a_1$**
> > >
> > > | Method | 0 | 10000 | 20000 | 30000 | 40000 | 50000 | 60000 |
> > > | --- | --- | --- | --- | --- | --- | --- | --- |
> > > | SARSA  | -0.590 | -0.210 | -0.123 | -0.088 | -0.068 | -0.054 | -0.045 |
> > > | Q-learning  | -0.590 | -0.058 | 0.055 | 0.093 | 0.107 | 0.113 | 0.118 |
> > > | Linear  | -0.590 | -0.070 | **0.025** | **0.044** | **0.041** | 0.032 | 0.019 |
> > > | Exp1  | -0.590 | -0.138 | -0.080 | -0.066 | -0.058 | -0.051 | -0.044 |
> > > | Exp2  | **-0.590** | **-0.056** | 0.057 | 0.093 | 0.109 | 0.114 | 0.110 |
> > > | Sigmoid  | -0.590 | -0.064 | 0.047 | 0.067 | 0.054 | **0.030** | **0.009** |

---

### Official Review · Reviewer_Q7Mh · 2025-03-15

**Overall Recommendation:** 3

**Summary:**

The manuscript proposes a gradual transition from a Bellman optimality operator to a Bellman operator by using a linearly annealed parameter to blend two Q-target estimates. I have noticed that a previously published paper introduced a BEE operator that similarly combines an exploitation-based update from historical best actions with an exploration-based update from the current policy using a λ parameter. Although the two works target different bias issues—overestimation in this manuscript versus underestimation in the earlier work—their underlying methods and even choices of algorithm backbone show striking similarities. Moreover, the manuscript does not discuss or reference this earlier work, making it challenging to distinguish its unique contributions.

**Claims And Evidence:**

Yes.

**Essential References Not Discussed:**

The manuscript proposes a gradual transition from the Bellman optimality operator to the Bellman operator by linearly annealing a parameter over time, thereby blending two Q-target estimates to accelerate learning while mitigating overestimation bias. The BEE paper[1], on the other hand, introduces an operator that linearly combines two different Bellman updates: one that exploits historically best-performing actions and another that uses the current policy for exploration, with the balance controlled by a trade-off parameter (λ). Although one work primarily focuses on reducing overestimation and the other on addressing underestimation in later training stages, both share a very similar underlying motivation—improving the accuracy of Q-value estimates within off-policy actor-critic frameworks by fusing two complementary update strategies. Moreover, the algorithmic implementation in both works is remarkably similar: each computes two separate Q-targets and then combines them using a parameter that is scheduled over time (BEE discussed some different scheduling mechanisms in its appendix), and they even make unexpectedly similar choices in terms of the algorithm backbone and experimental settings. This striking resemblance in both method and experimental design raises serious concerns regarding the originality of the manuscript, especially as it does not reference or discuss the BEE paper, leaving readers without a clear understanding of how the contributions of the manuscript differ or extend the existing work.

In light of these concerns, I recommend that the authors provide a detailed comparative discussion to clarify the unique contributions of their work. Specifically, they should explicitly highlight any theoretical or practical refinements that distinguish their approach from the one presented in the BEE paper. Without such a discussion, the high degree of overlap in core methodology, and even backbone selection significantly undermines the manuscript’s originality and its contribution to the field.

[1] Ji, T., Luo, Y., Sun, F., Zhan, X., Zhang, J., & Xu, H. (2024, July). Seizing Serendipity: Exploiting the Value of Past Success in Off-Policy Actor-Critic. In International Conference on Machine Learning (pp. 21672-21718). PMLR.

**Experimental Designs Or Analyses:**

Yes.

**Methods And Evaluation Criteria:**

Yes.

**Other Comments Or Suggestions:**

The main contribution of this manuscript could be considered as building upon the BEE paper by designing a more adaptive and intelligent trade-off mechanism for the $\lambda$ (or the $w$ parameter in this manuscript), allowing it to evolve naturally throughout the training process. If the manuscript seriously acknowledges this connection and revises its contribution accordingly, it would be a solid and acceptable work. Given that the experimental results are well-executed and provide strong empirical support, this refinement could make the contribution even more valuable.

**Other Strengths And Weaknesses:**

The experimental results are well-executed and provide strong empirical support.

**Questions For Authors:**

I find it lacks an overview table of the hyperparameters.

**Relation To Broader Scientific Literature:**

See the next part.

**Theoretical Claims:**

Yes.

---

> ### Author Rebuttal · Authors · 2025-03-31
>
> Thank you very much for pointing out that interesting and important prior work.
>
> BEE and AQL share a similarity in that both utilize the Bellman optimality operator based on in-sample maximization as well as the Bellman (expectation) operator. As the reviewer correctly noted, the key difference lies in the motivation. While BEE focuses on addressing underestimation, our approach emphasizes the potential benefits of overestimation in promoting learning, as supported by our preliminary experiments. Moreover, our method introduces a scheduling strategy that gradually decays optimality over time—a scheduling not explored in BEE.
>
> Additionally, unlike BEE, which combines the outputs of two separate Q-functions using different operators, our approach transitions gradually from one operator to the other. This allows us to maintain a single Q-function throughout training, simplifying the overall design.
>
> Therefore, our method can be seen as a simplification of BEE that introduces a novel motivation and scheduling mechanism to decay optimality, while relying on a single Q-function.
>
> We agree that the relationship with BEE is highly relevant, and we will revise our camera-ready version to explicitly discuss this connection and clarify our contributions accordingly.
>
> The hyperparameters used in our experiments are listed in Appendix B. To further improve clarity, we will also include an overview table summarizing them in the camera-ready version.
>
> Once again, we sincerely appreciate the valuable feedback. We hope that the inclusion of this discussion in the paper addresses your concerns. If not, we would greatly appreciate it if you could let us know what further explanations or revisions would be needed to improve the score.

---

> > ### Comment · Reviewer_Q7Mh · 2025-04-03
> >
> > Thank you for clarifying the connections and distinctions between your work and BEE, particularly regarding the scheduling mechanism and single Q-function approach. It is crucial that you thoroughly discuss BEE in the camera-ready version, given its importance as a key prior work. Highlighting both the similarities and differences, as well as providing an overview of the hyperparameters, should address my concerns. I look forward to seeing these improvements, and **based on these promised improvements, I have raised my score**.

---

> > > ### Author Response · Authors · 2025-04-08
> > >
> > > We sincerely appreciate your comment and for raising your score. We promise to include a thorough discussion of BEE and a table of hyperparameters in the camera-ready version.

---

### Official Review · Reviewer_dvAE · 2025-03-20

**Overall Recommendation:** 3

**Summary:**

This paper proposes to modify the Q-learning update in SAC and TD3 with an expectile loss like IQL, where the crucial proposition is to anneal the value of expectile $\tau$ from values close to 1 (representing max Q update) to 0.5 (representing SARSA update). The paper claims that "overestimation" in early stages of learning is beneficial for exploration and is shown to accelerate learning. However, at the end of training, reducing the overestimation bias is claimed to be desirable.

**Claims And Evidence:**

There is no theoretical or empirical evidence of the two key hypotheses:
1. why annealing should be done at all, as opposed to either of the two objectives. For instance, if max Q leads to overestimation, then one could use methods to reduce overestimation. Why do this specific kind of annealing to reduce overestimation? Similarly, if SARSA update leads to suboptimal or slow learning, why is it desirable to do it at all at the end of training?
2. how early max Q update leads to better exploration. There is no experiment that validates this claim.

Therefore, while the IQL-inspired method performs well in the experiments, the motivation behind the two key ideas of IQL-inspiration and annealing seem to be made up. Ideally, these ideas should be empirically validated to clearly delineate where exactly the gain is coming from.

----

Furthermore, in Section 3.2, the overestimation due to Q-learning is explored, but it is not clear whether this overestimation would change the optimal policy to the extent that one would prefer the Bellman operator instead of the Bellman optimality operator. In fact, we care about speed of convergence and optimality, and the overestimation should be self-resolved due to exploration in online RL (as mentioned in the paper) — so, it is not clear why one would not just choose to do max-Q like RL altogether. There needs to be a clear justification of the idea of balancing between max-Q and SARSA-style update.

----

The effectiveness of annealing in Section 4.3 is incorrectly claimed. In Table 2, the difference between Fixed (0.7) and Annealed (0.9) is so small that one might just prefer the fixed IQL method instead of the added complexity of annealing. Concretely, if we were to run two experiments: one with annealing from various values and one with fixed with various values, we would arrive at almost the same optimal result. Then, there is no meaningful benefit of the added complexity of annealing.

**Essential References Not Discussed:**

N/A

**Experimental Designs Or Analyses:**

1. The experiments do not justify the key problem that is present that is solved by the annealing. While there is a clear benefit offered due to the proposed method in the experiments, the source of this improvement is not well motivated or empirically justified.
2. The empirical result of Fixed (0.7) is quite strong, and it would be worthwhile to explore why that works as well. Annealing adds another layer of complexity that achieves the same effect as Fixed (0.7), while complicating the analysis to find the core reason for improvement.

**Methods And Evaluation Criteria:**

No clear justification for why a fixed $\tau$ is not enough and why one needs to do annealing to balance the max-Q and SARSA objectives, instead of a weighted balancing.

**Other Comments Or Suggestions:**

N/A

**Other Strengths And Weaknesses:**

## Strengths
- The observed gains in the experiments are statistically significant.

## Weaknesses
- The reasons for the gains are not attributed and motivated convincingly.

**Questions For Authors:**

- L268: "Overestimation increases the chance of the agent selecting overestimated actions, correcting them in the process, and thus broadening the range of actions tried."
How does it broaden the range of actions tried?

**Relation To Broader Scientific Literature:**

The idea extends IQL algorithm from offline RL to online RL, which already results in good results: Fixed (0.7). The addition of annealing does not bring any further statistically significant improvements.

**Theoretical Claims:**

1. The paper lacks a strong theoretical justification for the problem that annealing solves, and why annealing is a sound method to solve this problem.
2. There is no convergence or improvement proof that shows that the proposed change in the algorithm is a valid change that preserves stability and leads to better stability or convergence speed.

---

> ### Author Rebuttal · Authors · 2025-03-31
>
> Thank you very much for the highly valuable feedback. The concerns are comprehensively covered in the "Claims and Evidence" section. Below, we provide responses to each of the specific points mentioned there.
>
> **1. Why use annealing between max-Q and SARSA? If max-Q causes overestimation, why not reduce it directly? And if SARSA is suboptimal or slow, why use it at all?**
>
> Max-Q introduces an overestimation bias, but as discussed later, this can encourage exploration. Therefore, reducing this bias directly may not always be desirable.
>
> The reviewer asks, “If SARSA leads to suboptimal or slow learning, why use it at the end of training?” However, we emphasize that SARSA’s value estimates are not suboptimal. As an on-policy method, SARSA avoids the distribution mismatch between the actions used for target estimation and those actually taken. Furthermore, since SARSA does not involve Max, it introduces less bias. Although it may learn more slowly, it provides more accurate value estimates, as shown in Section 3.2.
>
> **2. How early max Q update leads to better exploration. There is no experiment that validates this claim.**
>
> We agree and appreciate this point. To address it, we measured policy entropy during early training. The table below shows average entropy over DMC tasks for SAC (Fixed (0.5), (0.7), and (0.9)), corresponding to SAC with expectile loss with $\tau = 0.5, 0.7, 0.9$. As shown in Figure 12, these methods exhibit overestimation bias in the order: Fixed (0.9) > Fixed (0.7) > SAC. The entropy follows the same trend, suggesting higher bias leads to broader exploration.
>
> This supports the intuitive hypothesis that overestimation can lead to suboptimal actions due to inflated Q-values, promoting  exploration. Prior work (e.g., Section 3 of [1]) also demonstrates that overestimation can be beneficial in tasks where exploration is important, while it degrades performance in tasks where exploration is undesirable. These results support the claim that overestimation bias promotes exploration.
>
> We will include these points in the camera-ready version to clarify the link between exploration and overestimation.
>
>
> [1] Q. Lan, Y. Pan, A. Fyshe, and M. White. Maxmin q-learning: Controlling the estimation bias of q-learning. In ICLR, 2020.
>
> **Table**: The average policy entropy in DM Control tasks. 15k steps corresponds to shortly after training begins at 10k steps.These results suggest that a large $\tau$ in the early stages of training increases policy entropy and promotes exploration.
> | Method           | 15k steps        | 100k steps       | 200k steps       |
> |------------------|------------------|------------------|------------------|
> | SAC (Fixed (0.5))  | 6.78 ± 0.30      | 6.23 ± 0.39      | 5.97 ± 0.38      |
> | Fixed (0.7)      | 7.41 ± 0.36      | 6.40 ± 0.40      | 6.11 ± 0.38      |
> | Fixed (0.9)      | 8.28 ± 0.44      | 6.94 ± 0.39      | 6.52 ± 0.38      |
>
>
> **3. Why not just use max-Q throughout, since overestimation can be mitigated by exploration? Why is it necessary to balance max-Q and SARSA-style updates?**
>
> While exploration can reduce overestimation, it doesn't eliminate it entirely. As shown in Figures 2 (right) and 12, higher $\tau$ values lead to greater final bias, exceeding that of SAC, a SARSA-style method. AQ-SAC starts with $\tau = 0.9$ to encourage exploration but gradually reduces bias to SAC's level, highlighting the benefit of balancing max-Q and SARSA updates. Fixed $\tau$ settings (e.g., 0.9, 0.95) perform poorly (Table 2), indicating that persistent overestimation harms optimality. While overestimation tends to diminish with more data, prior studies [1,2,3] have shown that actively suppressing it can significantly enhance sample efficiency. Balancing both perspectives, our method offers a natural compromise.
>
> [2] Deep reinforcement learning with double Q-Learning. Hado van Hasselt, Arthur Guez, David Silver. AAAI 2016.
>
> [3] Addressing Function Approximation Error in Actor-Critic Methods. Scott Fujimoto, Herke van Hoof, David Meger. ICML 2018.
>
>
>
> **4. If a well-chosen fixed τ performs similarly to annealing, is the added complexity of annealing really necessary?**
>
>  We acknowledge that a carefully selected fixed $\tau$ can yield good performance. However, as shown in Table 2, performance becomes highly sensitive to $\tau$ when it increases beyond a certain threshold, and performance can deteriorate significantly due to the increased bias.
>
> In RL tasks where sample efficiency is critical and many hyperparameters are inherently delicate, it is often impractical to finely tune all hyperparameters. In this context, our annealing strategy, which provides robustness against hyperparameter sensitivity, is a meaningful advantage.
>
> Once again, we sincerely appreciate the thoughtful and constructive feedback. It has helped us clarify and strengthen the motivation and evidence behind our proposed method. Please feel free to let us know if you have any further questions or concerns.

---

> > ### Comment · Reviewer_dvAE · 2025-04-06
> >
> > I appreciate the added experiment that quantifies the "better exploration due to max-Q" with high entropy policies in the early stages of training. I think this is central to the claim of the paper and justifies both why max-Q is used at all and why annealing should be done. I hope the authors will incorporate this experiment and explanation better in their text. I have updated my review accordingly.

---

> > > ### Author Response · Authors · 2025-04-08
> > >
> > > Thank you for your valuable feedback and for updating your score. We will include the results of the additional experiments along with explanatory text in the camera-ready version. Once again, we truly appreciate the time you took to provide such insightful feedback.

---

### Official Review · Reviewer_VRtg · 2025-03-20

**Overall Recommendation:** 4

**Summary:**

This paper proposed Annealed Q-learning to gradually transition from the Bellman optimality operator to the Bellman operator, leveraging early optimistic exploration while reducing overestimation bias during convergence. The approach is introduced via an illustrative example and performance is verified on a selection of continuous control tasks, where it performs favorably compared to established baseline algorithms. Overall, the approach is relatively simple and performs well on the tasks considered.

**Claims And Evidence:**

The approach is fairly straight-forward, introduced via an illustrative example and validated on continuous control benchmarks, with insightful ablations in the appendix. The provided evidence supports the claims, while it could be nice to run individual tasks to convergence to further strengthen the paper.

**Essential References Not Discussed:**

- Line 243: please revise the naming to avoid collision with Amortized Q-learning (AQL) in [1]. AQ-XXX should be fine, so maybe go for AQ-L to avoid an exact match with AQL?
- Please also check [2] for an alternative approach to reducing overestimation bias in Q-learning for (discretized) continuous control

**Additional references:**

[1] T. Van de Wielde, et al. "Q-learning in enormous action spaces via amortized approximate maximization." arXiv, 2020.

[2] D. Ireland, and Giovanni Montana. "Revalued: Regularised ensemble value-decomposition for factorisable markov decision processes." ICLR, 2024.

**Experimental Designs Or Analyses:**

The experimental validation is in line with prior works and shows the method comparing favorably. The following would further strengthen the paper:
- Longer runs on Humanoid-Walk/Run would be helpful to actually judge convergence.
- Have you tried to run experiments on DMC Dog tasks? Would be interesting to see regarding scalability?
- It would be interesting to see how the AQ-XXX agents compare to their XXX counterpart in action-penalized environments. Would they converge to local cost-minimization optima more quickly and get stuck there?
- Figure 4: SAC has commonly struggled on DMC tasks and AQ-SAC’s improvements are great to see. Please also add a D4PG baseline, as this would provide a strong reference (original DMC reference agent).

**Methods And Evaluation Criteria:**

The evaluation criteria are in line with the literature. Potential areas for improvement are mentioned below.

**Other Comments Or Suggestions:**

- Line 409: a potential counter-argument to “accurate action selection becomes difficult” is that highly discretized control appeared to be sufficient for the tasks evaluated in this paper
- Figure 5, caption: locomotion tasks —> manipulation tasks

**Other Strengths And Weaknesses:**

- Having an adaptive method for selecting the annealing constant would significantly strengthen the contribution

**Questions For Authors:**

- Have you thought about potential extensions towards “distributional RL” methods?

**Relation To Broader Scientific Literature:**

The general literature review is good, please see below for additional papers to consider.

**Theoretical Claims:**

The paper focuses more on quantitative validation without specific theoretical claims.

---

> ### Author Rebuttal · Authors · 2025-03-31
>
> Thank you very much for your insightful and helpful comments. Below, we address each of the points raised:
>
> **Additional experiments (Longer runs on Humanoid-Walk/Run, DMC Dog tasks, action penalty env, D4PG baseline):**
>
>  We appreciate the suggestion to conduct additional experiments. Due to limited computational resources, it is difficult to perform all the experiments; however, we have started running several of them and will share the results once they are complete.
>
> **Related work [1, 2]:**
>
>  Thank you for pointing out the name collision with Amortized Q-learning. We will rename our method to AQ-L in the camera-ready version to avoid confusion. We also appreciate the pointer to [2], and we will add a discussion on this work as part of the related research on action discretization.
>
> **Potential extensions towards distributional RL:**
>
>  As in QR-DQN [3], estimating Q-values for each discretized expectile and progressively reducing the target expectile may improve the consistency of the loss function during training. We believe this is a promising direction for extending our approach.
>
> **Line 409 – Counter-argument to "accurate action selection becomes difficult":**
>
> While discretization can be sufficient in certain tasks, it may lead to reduced sample efficiency in high-dimensional settings. For example, in the humanoid-run task, our method achieves a return of 250 within 3M steps, whereas [2] requires over 5M steps to reach the same performance. We will incorporate this discussion, including the effectiveness of discretization in lower-dimensional tasks, into the camera-ready version.
>
> **Figure 5 caption — "locomotion tasks" → "manipulation tasks":**
>
>  Thank you for catching this error. We will correct it in the final version.
>
>
> Once again, we sincerely thank you for the constructive and valuable feedback. Please feel
> free to reach out if there are any further questions or points that require clarification.
>
> [1] T. Van de Wielde, et al. "Q-learning in enormous action spaces via amortized approximate maximization." arXiv, 2020.
>
> [2] D. Ireland, and Giovanni Montana. "Revalued: Regularised ensemble value-decomposition for factorisable markov decision processes." ICLR, 2024.
>
> [3] Will Dabney, Mark Rowland, Marc G Bellemare, and Rémi Munos. Distributional reinforcement learning with quantile regression. In Thirty-Second AAAI Conference on Artificial Intelligence, 2018.

---

> > ### Comment · Reviewer_VRtg · 2025-04-04
> >
> > Thank you very much for the detailed response and clarifications! The additional experiments would be very interesting to see, looking forward to the results!

---

> > > ### Author Response · Authors · 2025-04-08
> > >
> > > As an update, we would like to share our current results regarding the suggested experiment on “Longer runs on Humanoid-Walk/Run.” Specifically, we extended the training of the model reported in the paper (originally trained up to 3e6 steps) to 1e7 steps. The results, shown below in terms of moving average of returns, indicate that our method converges to better performance than the baseline.
> > >
> > > **humanoid-run**
> > >
> > > | Method   | 1M           | 2M            | 3M            | 4M            | 5M            | 6M            | 7M            | 8M            | 9M            | 10M           |
> > > |----------|--------------|---------------|---------------|---------------|---------------|---------------|---------------|---------------|---------------|----------------|
> > > | SAC      | 1.60 ± 0.17  | 27.30 ± 16.47 | 37.24 ± 22.35 | 57.99 ± 25.44 | 70.20 ± 27.78 | 84.39 ± 33.35 | 82.67 ± 31.48 | 83.52 ± 32.69 | 95.03 ± 38.01 | 94.93 ± 36.75  |
> > > | AQ-SAC   | 106.46 ± 3.48 | 160.82 ± 4.53 | 215.35 ± 9.29 | 239.70 ± 15.75 | 267.69 ± 15.43 | 278.06 ± 20.66 | 236.32 ± 19.84 | 276.40 ± 16.42 | 279.17 ± 20.57 | 257.80 ± 25.77 |
> > >
> > > **humanoid-walk**
> > >
> > > | Method   | 1M            | 2M            | 3M            | 4M            | 5M            | 6M            | 7M            | 8M            | 9M            | 10M           |
> > > |----------|---------------|---------------|---------------|---------------|---------------|---------------|---------------|---------------|---------------|----------------|
> > > | SAC      | 120.20 ± 57.46 | 263.49 ± 80.94 | 417.96 ± 90.66 | 481.46 ± 103.48 | 561.67 ± 93.80 | 558.46 ± 92.13 | 575.77 ± 93.36 | 557.24 ± 91.89 | 569.29 ± 92.65 | 590.65 ± 97.36  |
> > > | AQ-SAC   | 381.65 ± 10.92 | 490.76 ± 15.64 | 620.55 ± 29.14 | 612.33 ± 24.15 | 655.15 ± 40.10 | 717.23 ± 19.04 | 719.35 ± 27.30 | 691.05 ± 23.90 | 688.10 ± 31.19 | 715.25 ± 19.53  |
> > >
> > > We believe these results are promising, especially given the lightweight setup with a batch size of 256 and a two-layer neural network with 256 units per layer. We expect that increasing the model capacity or using larger batch sizes could further enhance the performance on these challenging tasks.
> > >
> > > We will continue working on the other suggested experiments as well. Once again, we truly appreciate your valuable feedback and the time you have taken to review our work.

---

### Decision · Program_Chairs · 2025-05-01

**Decision:**

Accept (poster)

**Comment:**

This paper studies the effective integration of the Bellman optimality operator into actor-critic algorithms for online reinforcement learning in continuous action spaces, aiming to balance the fast but biased Q-learning with the stable but slower SARSA-based updates. To address this trade-off, the study introduces an annealing strategy that gradually shifts from the Bellman optimality operator to the standard Bellman operator, using expectile loss to enable smooth interpolation. The proposed Annealed Q-learning (AQL) is further implemented with TD3 and SAC to experimentally improve sample efficiency while mitigating overestimation bias.

In general, the proposed method appears to be simple yet effective. While some reviewers raised concerns about the lack of theoretical justification and comparisons to closely related work (notably the BEE paper), the authors addressed these concerns in their rebuttal by clarifying distinctions, extending discussions on prior work, and promising additional empirical comparisons in the camera-ready version. Furthermore, as acknowledged by reviewers, the empirical evaluation appears to be solid and thorough, demonstrating AQL’s effectiveness across a range of standard benchmarks. Overall, this paper introduces a timely and practically useful contribution to the deep RL research community.